# Single-cell entropy for accurate estimation of differentiation potency from a cell's transcriptome

Andrew E. Teschendorff[1,2,3] & Tariq Enver[3]

The ability to quantify differentiation potential of single cells is a task of critical importance. Here we demonstrate, using over 7,000 single-cell RNA-Seq profiles, that differentiation potency of a single cell can be approximated by computing the signalling promiscuity, or entropy, of a cell's transcriptome in the context of an interaction network, without the need for feature selection. We show that signalling entropy provides a more accurate and robust potency estimate than other entropy-based measures, driven in part by a subtle positive correlation between the transcriptome and connectome. Signalling entropy identifies known cell subpopulations of varying potency and drug resistant cancer stem-cell phenotypes, including those derived from circulating tumour cells. It further reveals that expression heterogeneity within single-cell populations is regulated. In summary, signalling entropy allows *in silico* estimation of the differentiation potency and plasticity of single cells and bulk samples, providing a means to identify normal and cancer stem-cell phenotypes.

[1] CAS Key Laboratory of Computational Biology, CAS-MPG Partner Institute for Computational Biology, Shanghai Institute for Biological Sciences, 320 Yue Yang Road, Shanghai 200031, China. [2] Department of Women's Cancer, University College London, 74 Huntley Street, London WC1E 6AU, UK. [3] UCL Cancer Institute, Paul O'Gorman Building, University College London, 72 Huntley Street, London WC1E 6BT, UK. Correspondence and requests for materials should be addressed to A.E.T. (email: a.teschendorff@ucl.ac.uk).

One of the most important tasks in single-cell RNA-sequencing studies is the identification and quantification of 'intercellular transcriptomic heterogeneity', that is, variation between the transcriptomes of single cells that is of biological relevance[1–4]. Although some of the observed intercellular transcriptomic variation represents stochastic noise, a substantial component has been shown to be of functional importance[1,5–8]. Very often, this biologically relevant heterogeneity can be attributed to cells occupying states of different potency or plasticity. Thus, quantification of differentiation potency, or more generally functional plasticity, at the single-cell level is of paramount importance. However, currently there is no concrete theoretical and computational model for estimating such plasticity at the single-cell level.

Here we make significant progress towards addressing this challenge. We propose a very general model for estimating cellular plasticity. A key feature of this model is the computation of signalling entropy[9], which quantifies the degree of uncertainty, or promiscuity, of a cell's gene expression levels in the context of a cellular interaction network. In effect, signalling entropy uses the transcriptomic profile of a cell to quantify the relative activation levels of its molecular pathways, and more generally that of biological processes, as defined over an *a priori* specified protein interaction network. We show that signalling entropy provides an excellent and robust proxy to the differentiation potential of a cell in Waddington's epigenetic landscape[10], and further provides a framework in which to understand the overall differentiation potency and transcriptomic heterogeneity of a cell population in terms of single-cell potencies. Attesting to its general nature and broad applicability, we compute and validate signalling entropy in over 7,000 single cells of variable degrees of differentiation potency and phenotypic plasticity, including time-course differentiation data, neoplastic cells and circulating tumour cells (CTCs). This extends entropy concepts that we have previously demonstrated to work on bulk tissue data[9,11–13] to the single-cell level. On the basis of signalling entropy, we develop a novel algorithm called single-cell entropy (SCENT), which can be used to identify and quantify biologically relevant expression heterogeneity in single-cell populations, as well as to reconstruct cell-lineage trajectories from time-course data. In this regard, SCENT differs substantially from other single-cell algorithms like Monocle[14], MPath[15], SCUBA[16], Diffusion Pseudotime[17] or StemID[18], in that it uses single-cell entropy to independently order single cells in pseudo-time (that is, differentiation potency), without the need for feature selection or clustering.

## Results

**The signalling entropy framework**. A pluripotent cell (by definition endowed with the capacity to differentiate into effectively all major cell-lineages) does not express a preference for any particular lineage, thus requiring a similar basal activity of all lineage-specifying transcription factors[9,19]. Viewing a cell's choice to commit to a particular lineage as a probabilistic process, pluripotency can therefore be characterized by a state of high uncertainty, or entropy, because all lineage choices are equally likely (Fig. 1a). In contrast, for a differentiated cell, or for a cell committed to a particular lineage, signalling uncertainty/entropy is reduced, as this requires activation of a specific signalling pathway reflecting that lineage choice (Fig. 1a). Thus, a measure of global signalling entropy, if computable, could provide us with a relatively good proxy of a cell's overall differentiation potential. Here we propose that differentiation potential can be estimated *in silico* by integrating a cell's transcriptomic profile with a high quality protein–protein interaction (PPI) network to define a cell-specific probabilistic signalling process (in effect, a random

walk) on the network (Methods). Mathematically, this random walk is described by a stochastic matrix whose entries reflect the relative interaction probabilities. Underlying the construction of these probabilities is the assumption that two genes, which can interact at the protein level, are more likely to do so if both are highly expressed (Fig. 1a, Methods). Given this stochastic matrix, global signalling entropy is then computed as the entropy rate (abbreviated as SR) of this probabilistic signalling process on the network[20] (Fig. 1b, Methods), and can be thought of as quantifying the overall level of signalling promiscuity of biological processes within the network. In effect, this quantifies the efficiency, or speed, with which signalling can diffuse over the whole network, and therefore measures the number of separate biological processes which are in some sense 'active'. Since a committed, or differentiated cell, preferentially activates and deactivates specific processes (pathways) in the network, the expectation is that this would manifest itself as a lower entropy rate since signalling cannot diffuse to the regions of the network describing inactive processes.

**Signalling entropy approximates differentiation potency**. To test that signalling entropy correlates with differentiation potency, we first estimated it for 1,018 single-cell RNA-Seq profiles generated by Chu *et al.*[21], which included pluripotent human embryonic stem cells (hESCs) and hESC-derived progenitor cells representing the three main germ layers (endoderm, mesoderm and ectoderm) ('Chu *et al.* set', Supplementary Table 1, Methods). In detail, these were 374 cells from two hESC lines (H1 & H9), 173 neural progenitor cells (NPCs), 138 definite endoderm progenitors (DEPs), 105 endothelial cells (ECs) representing mesoderm derivatives, as well as 69 trophoblast cells (TB) and 148 human foreskin fibroblasts (HFFs). Confirming our hypothesis, pluripotent hESCs attained the highest signalling entropy values, followed by multipotent cells (NPCs, DEPs), and with less multipotent HFFs, TBs and ECs attaining the lowest values (Fig. 2a). Differences were highly statistically significant, with DEPs exhibiting significantly lower entropy values than hESCs (Wilcoxon rank-sum $P < 1e - 50$) (Fig. 2a). Likewise, TBs exhibited lower entropy than hESCs ($P < 1e - 50$), but higher than HFFs ($P < 1e - 7$) (Fig. 2a). Importantly, signalling entropy correlated very strongly with a pluripotency score obtained using a previously published pluripotency gene expression signature[22] (Spearman correlation = 0.91, $P < 1e - 500$, Fig. 2b, Methods). In all, signalling entropy provided a highly accurate discriminator of pluripotency versus non-pluripotency at the single-cell level (AUC = 0.96, Wilcoxon test $P < 1e - 300$, Fig. 2c). We note that in contrast with pluripotency expression signatures, this strong association with pluripotency was obtained without the need for any feature selection or training.

To further test the general validity and robustness of signalling entropy we computed it for scRNA-Seq profiles of 3,256 non-malignant cells derived from the microenvironment of 19 melanomas (Melanoma set[23], Supplementary Table 1). Cells profiled included T-cells, B-cells, natural killer (NK) cells, macrophages, fully differentiated ECs and cancer-associated fibroblasts (CAFs). For a given cell-type and individual, variation between single cells was substantial and similar to the variation seen between individuals (Supplementary Fig. 1). Mean entropy values however, were generally stable, showing little inter-individual variation, except for T-cells from 4 out of 15 patients, which exhibited a distinctively different distribution (Supplementary Fig. 1). To assess overall trends, we pooled the single-cell entropy data from all patients together, which confirmed that all lymphocytes (T-cells, B-cells and NK cells) had similar average signalling entropy values (Fig. 2d).

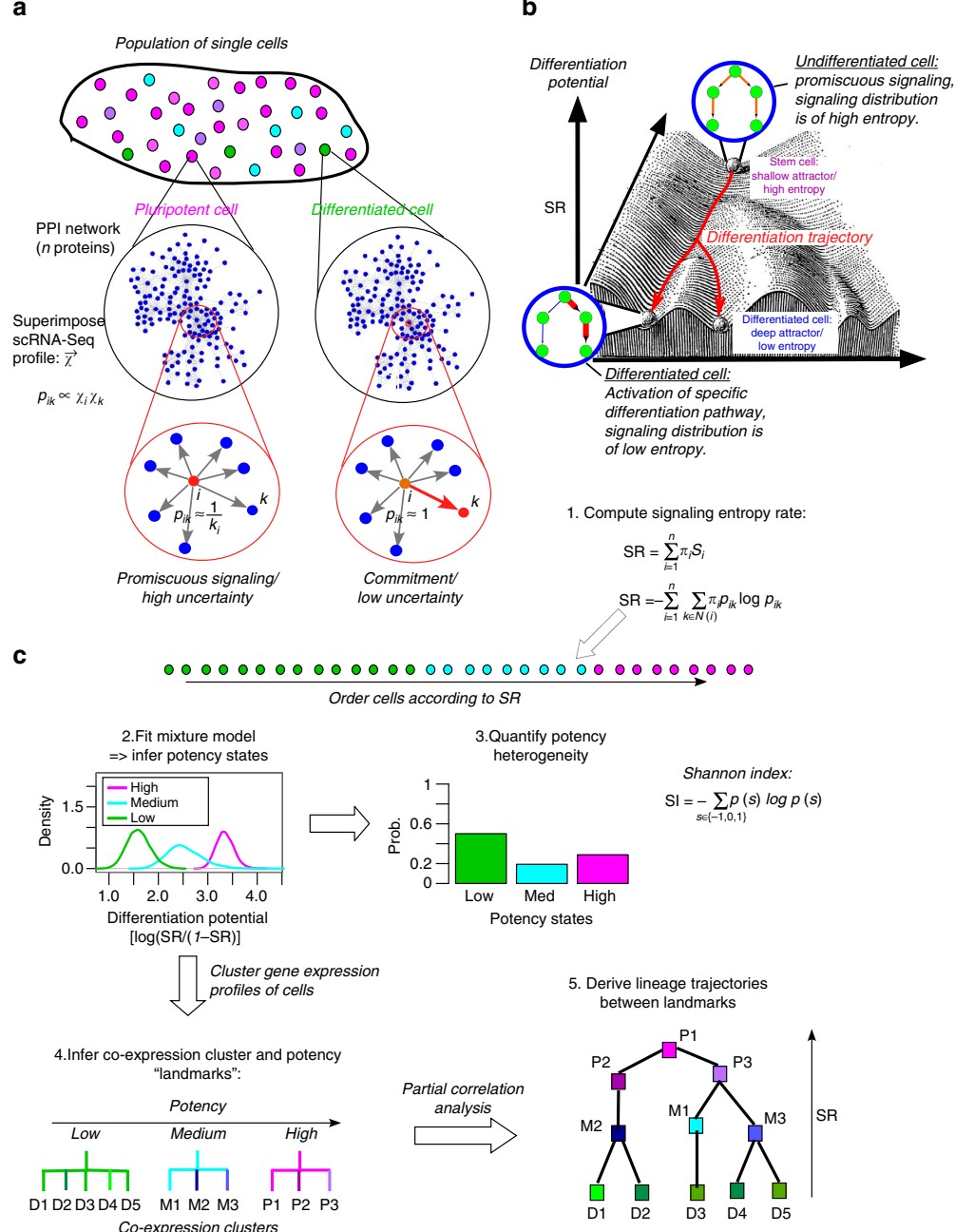

**Figure 1 | The single-cell entropy (SCENT) algorithm. (a)** Signalling entropy of single cells as a proxy to their differentiation potential in Waddington's landscape. Depicted on the left is a population of cells with cells occupying either a pluripotent (magenta), a progenitor (cyan) or a differentiated state (green). The potency state of each cell is determined by a complex function of the transcriptomic profile $\vec{x}$ of the cell. For a given interaction between proteins $i$ and $k$ in the network, signalling in a given cell occurs with a probability $p_{ik} \sim x_i x_k$, defining a stochastic matrix $P = (p_{ik})$. In a pluripotent state, there is high demand for phenotypic plasticity, and so promiscuous signalling proteins (that is, those of high connectivity) are highly expressed (red coloured node) with all major differentiation pathways kept at a similar basal activity level (grey edges). The probability of signalling between protein $i$ and $k$, $p_{ik}$, is therefore $1/k_i$ where $k_i$ is the connectivity of protein $i$ in the network. Thus the local signalling entropy around node $i$ is maximal. In a differentiated state, commitment to a specific lineage (activation of a specific signalling pathway shown by red coloured node) means that most $p_{ij} \sim 0$, except when $j = k$, so that $p_{ik} \sim 1$. Thus, local signalling entropy around node $i$ is close to zero. **(b)** Estimation of signalling entropy. An overall measure of signalling promiscuity of the cell is given mathematically by the signalling entropy rate (SR), which is a weighted average of local signaling entropies $S_i$ over all the genes/proteins in the network, with weights specified by $\pi$ (the steady-state probability satisfying $\pi P = \pi$). It is proposed that SR provides a proxy to the elevation in Waddington's landscape, quantifying differentiation potential of cells (i.e the number of accessible cell-fates within a given lineage). **(c)** Quantification of intercellular heterogeneity and reconstruction of lineage trajectories. Estimation of signalling entropy at the single-cell level across a population of cells, allows the distribution of potency states in the population to be determined through Bayes mixture modelling which infers the optimal number of potency states. From this, the heterogeneity of potency states in a cell population is computed using Shannon's Index. To infer lineage trajectories, SCENT uses a clustering algorithm over dimensionally reduced scRNA-Seq profiles to infer co-expression clusters of cells. Dual assignment of cells to a potency state and co-expression cluster allows the identification of landmarks as bi-clusters in potency-coexpression space. Finally, partial correlations between the expression profiles of the landmarks are used to infer a lineage trajectory network diagram linking cell clusters according to expression similarity, with their height or elevation determined by their potency (signalling entropy).

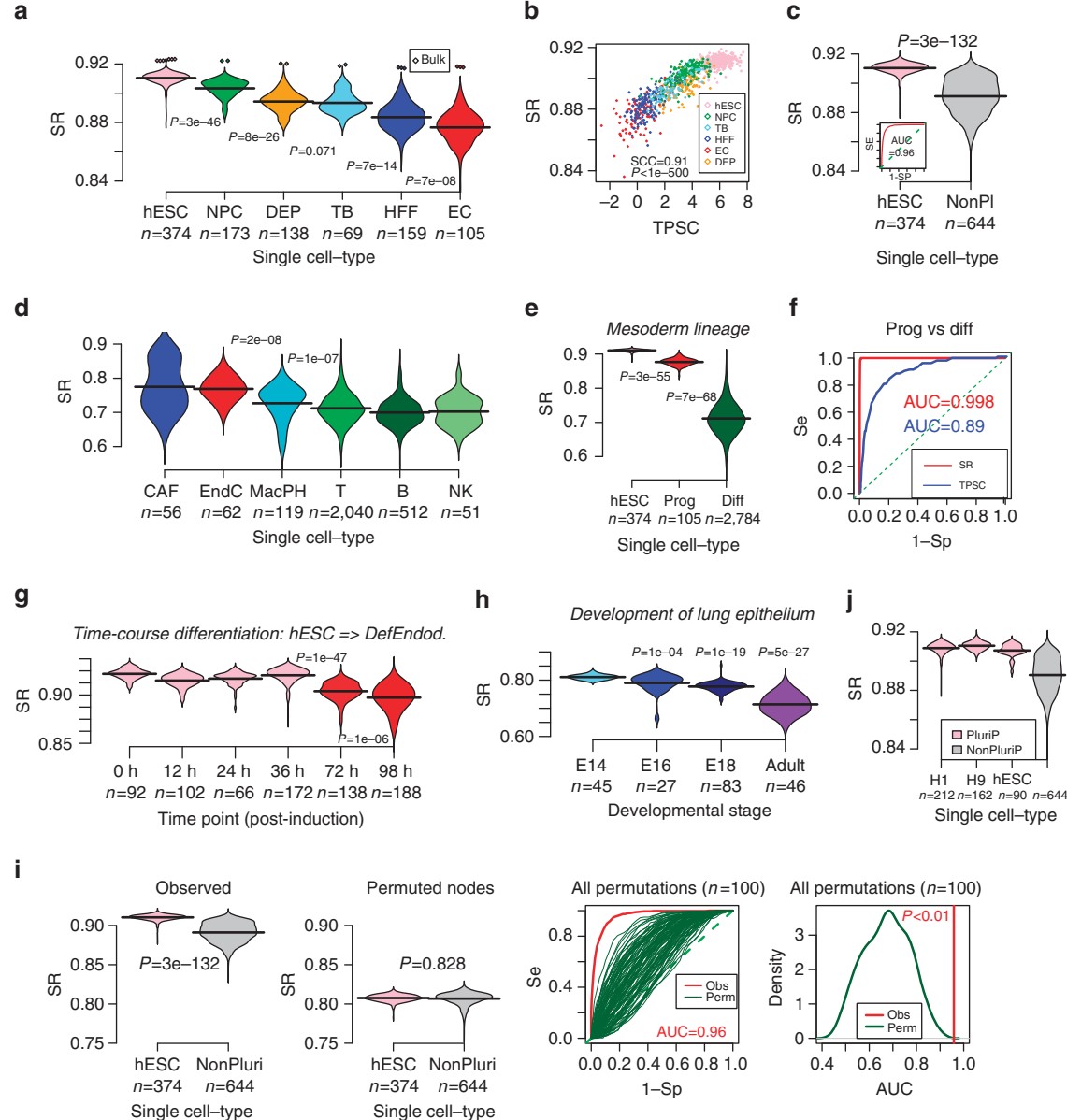

**Figure 2 | Signalling entropy correlates with differentiation potency of single cells.** (**a**) Violin plots of the signalling entropy (SR) against cell-type (DEP, definite endoderm progenitors; EC, endothelial cells (mesoderm progenitor derivatives); HFF, human foreskin fibroblasts; hESC, human embryonic stem cells; NPC, neural progenitor cells; TB, trophoblast cells). Number of single cells in each class is indicated. Total number is 1,018. Wilcoxon rank-sum test *P* values between each cell-type (ranked in decreasing order of SR) are given. Diamond shaped data points correspond to the matched bulk samples. (**b**) Scatterplot of the signalling entropy (SR, *y*-axis) against an independent mRNA expression based pluripotency score (TPSC, *x*-axis) for all 1,018 single cells. Cell-type is indicated by colour. Spearman Correlation Coefficient (SCC) and associated *P* value are given. (**c**) Violin plot comparing the signalling entropy (SR) between the hESCs and all other (non-pluripotent) cells. *P* value is from a Wilcoxon rank-sum test. Inlet figure is the associated ROC curve, which includes the AUC value. (**d**) Violin plot of signalling entropy (SR) values for non-malignant single cells found in the microenvironment of melanomas. Number of single cells of each cell-type are given (B, B-cells; CAF, cancer-associated fibroblasts; EndC, endothelial cells; MacPH, macrophages; NK, natural killer cells; T, T-cells). Wilcoxon rank-sum test *P* values between EndC and MacPH, and between MacPH and all lymphocytes are given. (**e**) Signalling entropy (SR) as a function of differentiation stage within the mesoderm lineage. Differentiation stages include hESCs (pluripotent), mesoderm progenitors of endothelial cells (multipotent) and differentiated endothelial and white blood cells. Wilcoxon rank-sum test *P* values between successive stages are given. (**f**) ROC curves and AUC values for discriminating the progenitor and differentiated cells within the mesoderm lineage for signalling entropy (SR) and the *t*-test pluripotency score (TPSC). (**g**) Signalling entropy (SR, *y*-axis) as a function of time in a single-cell time course differentiation experiment, starting from hESCs at time = 0 h (time of differentiation induction) into definite endoderm (which occurs from 72 h onwards). Number of single cells measured at each time point is given. Wilcoxon rank-sum test *P* values between the first four time points and 72 h, and between 72 and 98 h are given. (**h**) Signalling entropy (SR, *y*-axis) as a function of developmental stage in the differentiation of the distal mouse lung epithelium. Number of single cells measured at each stage is given. Wilcoxon rank-sum test *P* values between embryonic day 14 (E14) and all other stages are given. (**i**) Comparison of the SRs in **c**) (left panel) to the case where expression values are randomly reshuffled before computation of SR (middle panel). Right panels compare the corresponding ROC curves and AUC values. (**j**) As **c**, but now splitting the hESCs into cells from H1 and H9 lines, and including an additional independent set of 90 single hESCs profiled with a different NGS platform.

**Table 1 | Comparison of signalling entropy to SLICE and StemID as measures of differentiation potency in scRNA-Seq and bulk RNA-Seq data sets.**

| Data set | | Signalling entropy | SLICE | StemID |
|---|---|---|---|---|
| *scRNA-Seq* | | | | |
| Chu1 (Pl > NonPl) | P | 3e − 132 | ~1 | 3e − 58 |
| | AUC | **0.96** | <0.5 | 0.79 |
| Chu2 (0 h > 96 h) | P | 2e − 38 | 0.94 | 1e − 22 |
| | AUC | **0.97** | <0.5 | 0.86 |
| Trapnell (0 h > 72 h) | P | 6e − 9 | 0.0003 | 2e − 10 |
| | AUC | 0.74 | 0.65 | **0.75** |
| Treutlein (E14 > Adult) | P | 5e − 27 | 6e − 26 | 5e − 27 |
| | AUC | **1** | **0.998** | **1** |
| | | | | |
| *Bulk RNA-Seq* | | | | |
| Chu3 (Pl > NonPl) | P | 4e − 5 | 0.001 | 0.76 |
| | AUC | **0.99** | 0.90 | <0.5 |

Table lists one-tailed Wilcoxon rank-sum test *P* values and associated (one-tailed) AUCs, testing whether entropy is higher in the pluripotent or multipotent cells compared to the less potent cells in various scRNA-Seq and bulk RNA-Seq data sets. In Chu1, the comparison is between pluripotent (hESCs, $n = 374$, Pl) and non-pluripotent ($n = 644$, NonPl) single cells. In Chu2, the comparison is between hESCs (0 h, $n = 92$) and definite endoderm progenitors sampled 96 h later ($n = 188$). In Trapnell, the comparison is between human myoblasts (0 h, $n = 96$) and differentiated skeletal muscle cells (72 h, $n = 84$). In Treutlein, the comparison is between early lung progenitors (E14, $n = 45$) and mature alveolar cells ($n = 46$). In Chu3, the comparison is between bulk hESCs ($n = 7$) and non-pluripotent samples ($n = 12$). In bold-face we indicate the highest or relatively highest AUC values.

Intra-tumour macrophages, which are derived from monocytes, exhibited a marginally higher signalling entropy (Fig. 2d). The highest signalling entropy values were attained by ECs and CAFs (Fig. 2d), consistent with their known high phenotypic plasticity[24–27]. Importantly, the entropy values for all of these non-malignant differentiated cell-types were distinctively lower compared to those of hESCs and progenitor cells from Chu *et al.* (Fig. 2a,d), consistent with the fact that hESCs and progenitors have much higher differentiation potency. To test this formally, we compared hESCs, mesoderm progenitors, and terminally differentiated cells within the mesoderm lineage (which included all ECs and lymphocytes), which revealed a consistent decrease in signalling entropy between all three potency states (Wilcoxon rank test $P < 1e − 50$, Fig. 2e). Of note, signalling entropy could discriminate progenitor and differentiated cells better than the score derived from the pluripotency gene expression signature[22], attesting to its increased robustness as a general measure of differentiation potency (Fig. 2f, Supplementary Fig. 2).

Next, we assessed signalling entropy in the context of a time-course differentiation experiment, whereby hESCs were induced to differentiate into DEPs via the mesoendoderm intermediate[28]. scRNA-Seq for a total of 758 single cells, obtained at six time points, including origin, 12, 24, 36, 72 and 96 h post induction were available (Methods)[28]. We observed that single-cell entropies exhibited a particular large decrease only after 72 h (Fig. 2g), consistent with previous knowledge that differentiation into definite endoderm occurs around 3–4 days after induction[28]. To demonstrate the validity of signalling entropy in another species, we next considered a scRNA-Seq data of cells sampled at different embryonic stages in the development of the mouse lung epithelium[29] ('Treutlein set', Supplementary Table 1, Methods). Signalling entropy decreased continuously until adulthood in line with a gradual increase in differentiation (Fig. 2h). Moreover, at embryonic day 18, it could discriminate alveolar type cells from a recently discovered bipotent progenitor subgroup[29], albeit with marginal significance due to small cell numbers (Supplementary Fig. 3A).

To demonstrate the critical importance of the interaction network, we recomputed signalling entropy in the Chu and Treutlein data sets after randomly reshuffling gene expression values over the network (100 and 1,000 permutations, respectively). As expected, upon reshuffling, signalling entropy lost its power to discriminate pluripotent from non-pluripotent cells

(Fig. 2i), and did not exhibit a consistent decrease with developmental stage in Treutlein's set (Supplementary Fig. 3B).

**Robustness to choice of PPI network and NGS platform.** Given the importance of the PPI network, it is therefore equally important to verify that signalling entropy is robust to the choice of network. Results were largely unchanged using a different version of a PPI network (Supplementary Fig. 4). To test the robustness of signalling entropy across independent studies, we analysed scRNA-Seq data for an independent set of single-cell hESCs derived from the primary outgrowth of the inner cell mass ('hESC set'[30], Supplementary Table 1). Obtained signalling entropy values were most similar to those of single cells derived from the H1 and H9 hESC lines, confirming the robustness of signalling entropy across different studies and next-generation sequencing platforms (Fig. 2j, Supplementary Table 1).

**Comparison of signalling entropy to StemID and SLICE.** To further highlight the importance of the PPI network, we decided to compare Signalling Entropy to two other entropy-based potency measures, proposed as part of the StemID[18] and SLICE[31] algorithms, which we note do not use any network information. To provide an objective evaluation, we compared the entropy measures of single cells from well-separated differentiation stages, or by comparing start and end points in time-course differentiation experiments, as these cells ought to differ substantially in terms of potency. Adopting this strategy in four scRNA-Seq and one bulk RNA-Seq data set, we observed that signalling entropy was able to provide high discriminative power in each data set (Table 1). In contrast, we did not find StemID and SLICE to be as accurate or robust (Table 1).

**Correlation with potency is independent of cell-cycle phase.** A major source of variation in scRNA-Seq data is cell-cycle phase[23,32]. We explored the relation between signalling entropy and cell-cycle phase in a large scRNA-Seq data set encompassing 3,256 non-malignant and 1,257 cancer cells derived from the microenvironment of melanomas (Melanoma set[23], Supplementary Table 1). A cycling score for both G1-S and G2-M phases and for each cell was obtained using a validated procedure[23,32,33], and compared to signalling entropy, which revealed a strong yet highly non-linear correlation

(Supplementary Fig. 5). Specifically, we observed that cells with a low signalling entropy were never found in either the G1-S or G2-M phase (Supplementary Fig. 5). In contrast, cells with high signalling entropy could be found in either a cycling or non-cycling phase. These results are consistent with the view that cycling cells must increase expression of promiscuous signalling proteins and hence exhibit an increased signalling entropy. Thus, we next asked if signalling entropy correlates with potency when restricting to non-cycling cells. Using the Chu et al. data set, we observed that, although discrimination accuracies were reduced upon correction for cell-cycle phase, signalling entropy could still accurately classify pluripotent from non-pluripotent cell-types (AUC > 0.9, $P < 1e - 5$, Supplementary Fig. 6, Supplementary Table 2). Consistent with this (and now using both cycling and non-cycling cells), the correlation between signalling entropy and potency remained significant when adjusted for cell-cycle scores (Supplementary Table 2).

**Correlation of expression with degree partly drives potency.** To gain further biological insight into signalling entropy, we derived an approximation for signalling entropy in terms of the three-way correlation between the transcriptome, connectome and local signalling entropies (Methods). This approximation implies that if, on average, network hubs are more highly expressed than low-degree nodes and if they exhibit an increase in their local signalling entropy, then this should generally lead to a more efficient distribution of signalling over the network, and hence to an increased global signalling entropy[12]. We thus posited that in cells with a demand for high phenotypic plasticity (for example, pluripotent cells), hubs tend to be overexpressed and exhibit increased signalling promiscuity. Using scRNA-Seq data from Chu et al.[21], we were able to confirm a weak (Pearson correlation of ~0.2) but significant ($P < 1e - 50$) positive correlation of differential gene expression (between hESCs and multipotent cells) with connectivity (Supplementary Fig. 7A). Importantly, the differential local signalling entropy between hESCs and multipotent cells correlated more strongly with connectivity (Pearson correlation of ~0.64, $P < 1e - 100$, Supplementary Fig. 7A), thus confirming the notion that the increased SR in pluripotent cells is also driven by a more distributed signalling (that is, increased local entropy) at network hubs. To demonstrate that the Pearson correlation between transcriptome and connectome can be used to approximate signalling entropy (SR), we computed it for all 1,018 single cells in Chu et al., obtaining an excellent agreement with SR ($R^2 = 0.96$, Supplementary Fig. 7B), and hence also with potency (Supplementary Fig. 7C). However, we stress that this Pearson correlation approximation is not a substitute for SR, since the definition of SR includes the local signalling entropies (Fig. 1b), from which important biological information can be extracted. To demonstrate this, we ranked genes in the network according to their differential local signalling entropy (Methods) and performed gene set enrichment analysis (GSEA)[34] on the genes exhibiting the most significant increases in local entropy between pluripotent (hESCs) and multipotent cells. Top-ranked enriched biological terms included, besides stemness, genes implicated in mRNA splicing and encoding mitochondrial ribosomal proteins (Supplementary Table 3, Supplementary Data 1). This is consistent with recent studies demonstrating that mitochondrial activity influences the global transcription and splicing rate of cells[35–37], and that variations in such activity may influence stemness and differentiation[38–42]. Finally, we also point out that signalling entropy and its Pearson correlation approximation are not equivalent, as there exist networks where both measures yield very different answers (Methods). For instance, in networks where hubs are not connected to each other (unlike our PPI networks where hubs are generally connected to each other), a positive correlation could lead to a lower signalling entropy (Supplementary Fig. 7D).

**Quantifying single-cell expression heterogeneity with SCENT.** Given that signalling entropy correlates with differentiation potency, we used it to develop the SCENT algorithm (Fig. 1c). Briefly, SCENT uses the estimated single-cell entropies to infer the distribution of discrete potency states across the cell population (Fig. 1c, Methods). Thus, SCENT can be used to quantify expression heterogeneity at the level of potency. In addition, SCENT can be used to directly order single cells in pseudo-time[14] to facilitate reconstruction of lineage trajectories. A key feature of SCENT is the assignment of each cell to a unique potency state and co-expression cluster, which results in the identification of potency clusters (which we call 'landmarks'), through which lineage trajectories are then inferred (Methods).

We first tested SCENT on the scRNA-Seq data from Chu et al., which profiled pluripotent and multipotent cells (Supplementary Table 1). SCENT correctly predicted a parsimonious two-state model, with a high potency pluripotent state and a lower potency non-pluripotent progenitor-like state (Fig. 3a). Interestingly, a small fraction (~4%) of hESCs were deemed to be non-pluripotent cells (Fig. 3b), consistent with previous observations that pluripotent cell populations contain cells that are already primed for differentiation into specific lineages[5,6]. Supporting this, these non-pluripotent 'hESCs' exhibited lower cycling scores and higher expression levels of neural (HES1/SOX2) and mesoderm (PECAM1) stem-cell markers, compared to the pluripotent hESCs (Supplementary Fig. 8). Whereas all HFFs and ECs were deemed non-pluripotent, DEPs, TBs and NPCs exhibited mixed proportions, with NPCs exhibiting approximately equal numbers of pluripotent and non-pluripotent cells (Fig. 3b). Correspondingly, the Shannon index (SI), which quantifies the level of heterogeneity in potency, was highest for the NPC population (Fig. 3c). In total, SCENT predicted six co-expression clusters, which combined with the two potency states, resulted in a total of seven landmark clusters (Fig. 3d). These landmarks correlated very strongly with cell-type, with only NPCs being distributed across two landmarks of different potency (Fig. 3e). SCENT correctly inferred a lineage trajectory between the high potency NPC subpopulation and its lower potency counterpart, as well as a trajectory between hESCs and DEPs (Fig. 3f). The other cell-types exhibited lower entropies (Fig. 2b, Fig. 3f), and correspondingly did not exhibit a direct trajectory to hESCs, suggesting several intermediate states which were not sampled in this experiment.

To ascertain the biological significance of the two NPC subpopulations (Fig. 3b,e,f), we first verified that the NPCs deemed pluripotent did indeed have a higher pluripotency score (Supplementary Fig. 9A), as assessed using the independent pluripotency gene expression signature from Palmer et al.[22] We further reasoned that well-known transcription factors marking neural stem/progenitor cells, such as HES1, would be expressed at a much lower level in the NPCs deemed pluripotent compared to the non-pluripotent ones, since the latter are more likely to represent bona fide NPCs. Confirming this, NPCs with low HES1 expression exhibited higher differentiation potential than NPCs with high HES1 expression (Wilcoxon rank-sum test $P < 0.0001$, Fig. 3g). Similar results were evident for other neural progenitor/stem cell markers such as PAX6 and SOX2 (Supplementary Fig. 9B). Of note, NPCs expressing the lowest levels of PAX6, HES1 or SOX2 were generally always classified by SCENT into a pluripotent-like state (Fig. 3g, Supplementary

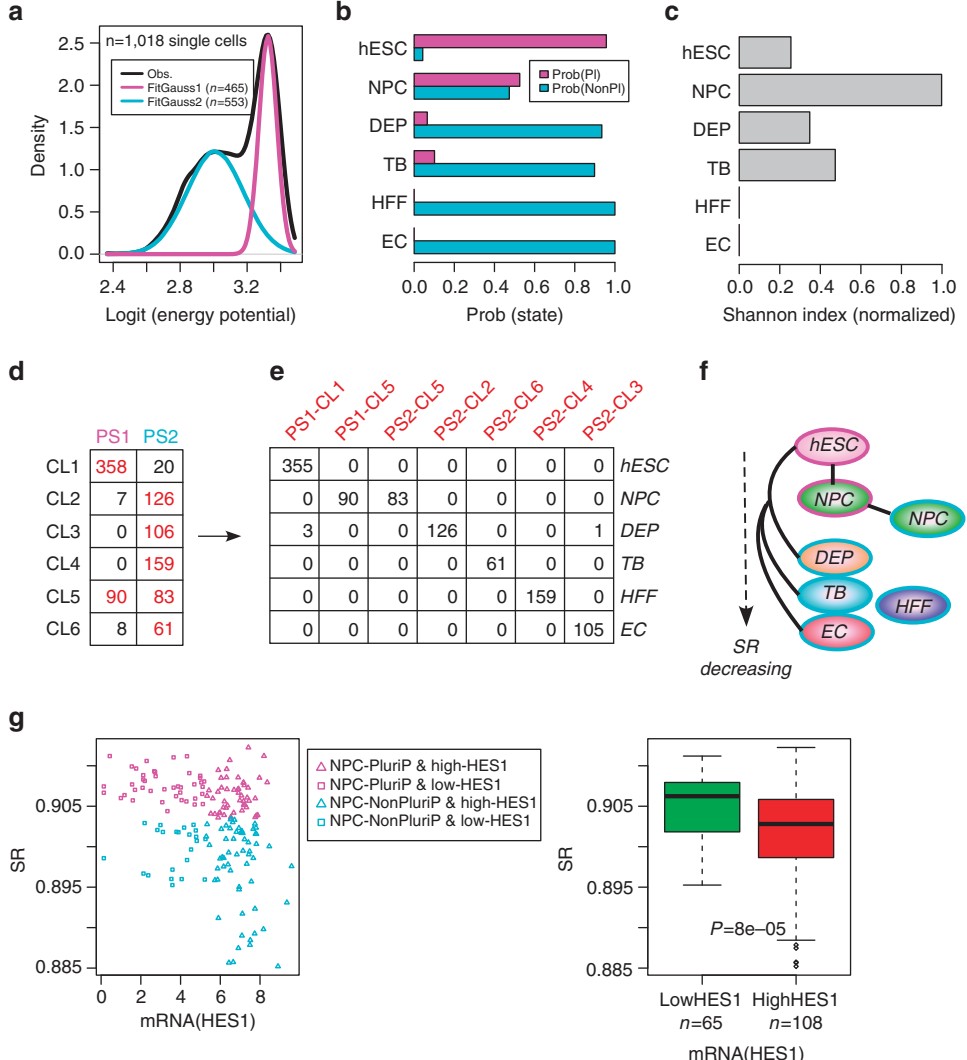

**Figure 3 | SCENT identifies single-cell subpopulations of biological significance.** (**a**) Fitted Gaussian mixture model to the signalling entropies of 1,018 single cells (scRNA-Seq data from Chu *et al.*) using a logit scale for the signalling entropies (x-axis, $\log_2[SR/(1-SR)]$). Bayesian Information Criterion predicted only two-states: a high energy/entropy pluripotent state (magenta-PS1) and a lower-energy non-pluripotent state (cyan-PS2). Number of cells categorized into each state is indicated in plot. (**b**) Barplot comparing, for each cell-type, the probability that a cell from this cell population is in the pluripotent (prob(PI)) or non-pluripotent state (probe(NonPI)). Cell-types include human embryonic stem cells (hESCs), neural progenitor cells (NPCs), definite endoderm progenitors (DEPs), trophoblast cells (TBs), human foreskin fibroblasts (HFFs) and endothelial cells (ECs). (**c**) Barplot of the corresponding Shannon Index for each cell-population type. (**d**) Distribution of single-cell numbers between inferred potency states and co-expression clusters, as predicted by SCENT. In brown, we indicate 'landmark clusters' which contain at least 5% of the total number of single cells. (**e**) Distribution of single-cell-types among the seven landmark clusters. (**f**) Inferred lineage trajectories between the seven landmarks which map to cell-types. Border colour indicates potency state: magenta = PS1, cyan = PS2. (**g**) Left panel: scatterplot of signalling entropy (SR) vs mRNA expression level of a neural stem/progenitor cell marker, HES1, for all NPCs. NPCs categorized as pluripotent are shown in magenta, NPCs categorized into a non-pluripotent state are shown in cyan. NPCs of high and low HES1 expression (as inferred using a partition-around-medoids algorithm with $k=2$) are indicated with triangles and squares, respectively. Right panel: corresponding boxplot comparing the differentiation potency (SR) of NPCs with low versus high HES1 expression. *P* value is from a one-tailed Wilcoxon rank-sum test.

Fig. 9B). Thus, these results indicate that SCENT provides a biologically meaningful characterization of intercellular transcriptomic heterogeneity.

**SCENT reconstructs lineage trajectories in differentiation.** We next tested SCENT in the context of a differentiation experiment of human myoblasts[14], involving skeletal muscle myoblasts which were first expanded under high mitogen conditions and later induced to differentiate by switching to a low serum medium (Trapnell *et al.* set, Supplementary Table 1). A total of 96 cells

were profiled with RNA-Seq at differentiation induction, as well as at 24 and 48 h after medium switch, with a remaining 84 cells profiled at 72 h. As expected, signalling entropy was highest in the myoblasts, with a switch to lower entropy occurring at 24 h (Fig. 4a). No further decrease in entropy was observed between 24 and 72 h, indicating that commitment of cells to become differentiated skeletal muscle cells already happens early in the differentiation process. Over the whole time course, SCENT predicted a total of 3 potency states, with a distribution consistent with the time of sampling (Fig. 4b). Cells sampled at differentiation induction were made up primarily of two

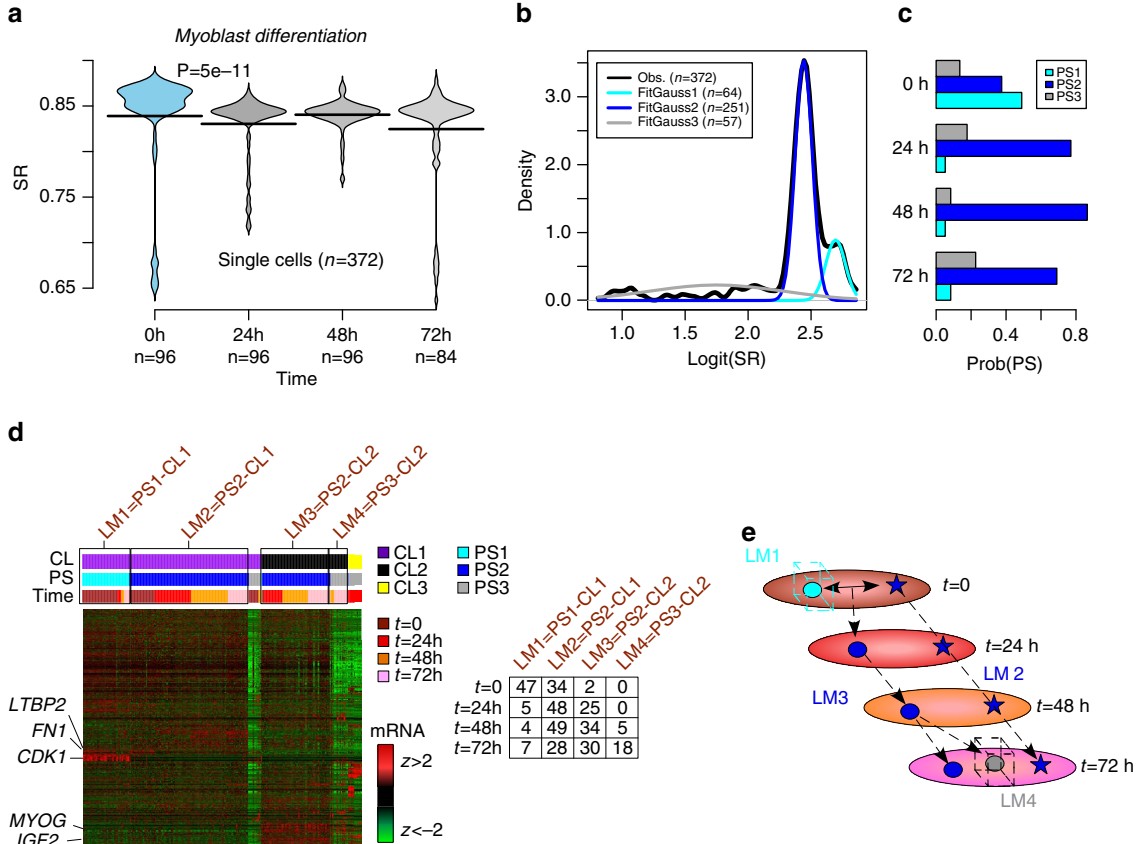

**Figure 4 | SCENT dissects distinct lineage trajectories in human myoblast differentiation.** (**a**) Signalling entropy (SR) versus time point (0 h, 24 h, 48 h, 72 h) for a total of 372 single cells, collected during a time-course differentiation experiment of human myoblasts (scRNA-Seq from Trapnell *et al.*). Violin plots show the density distribution of SR values at each time point. *P* value is from a one-tailed Wilcox rank-sum test comparing timepoint 0 h to 24 h. (**b**) SCENT Gaussian Model fit to SR values predicts three potency states (PS1, PS2, PS3). (**c**) Probability distribution of potency states at each timepoint. (**d**) Co-expression heatmap of highly variable genes obtained by SCENT predicting three main clusters. Single cells have been ordered, first by cluster, then by potency state and finally by their time of sampling, as indicated. Landmarks are indicated by rectangular boxes, and distribution of single cells across landmarks and time points is provided in table. Genes have been clustered using hierarchical clustering. Genes that are markers of the different landmarks have been highlighted. (**e**) Inferred lineage trajectories between landmarks. Diagram illustrates an inferred two-phase trajectory, with one trajectory describing myoblasts of high potency (t = 0, cyan circle) differentiating into skeletal muscle cells of intermediate potency (t = 24 and 48) (blue circles) and a mixture of terminally differentiated and intermediate potency skeletal muscle cells (t = 72) (grey and blue circle, respectively). A second trajectory/landmark describes a different cell-type (interstitial mesenchymal cells) whose intermediate potency state does not change during the time-course (blue stars).

potency states (Fig. 4c, PS1 & PS2), which differed in terms of *CDK1* expression, consistent with one subset (PS1) defining a highly proliferative subpopulation and with the rest (PS2) representing cells that have exited the cell cycle (Supplementary Fig. 10). In total, SCENT predicted four landmarks, with one landmark defining undifferentiated (t = 0) myoblasts of high potency (Fig. 4d). Another landmark of lower potency contained cells at all time points, with cells expressing markers of mesenchymal cells (for example, *PDFGRA* and *FN1/LTBP2*) (Fig. 4d). Cells from this landmark which were present at differentiation induction exhibited intermediate potency expressing low levels of *CDK1* (Supplementary Fig. 10, Fig. 4d), suggesting that these are 'contaminating' interstitial mesenchymal cells that were already present at the start of the time course, in line with previous observations[14,15]. Importantly, SCENT correctly predicts that the potency of all these mesenchymal cells in this landmark does not change during the time-course, consistent with the fact that these cells are not primed to differentiate into skeletal muscle cells, but which nevertheless aid the differentiation process[14,15]. Another landmark of intermediate potency predicted by SCENT defined a trajectory

made up of cells expressing high levels of myogenic markers (*MYOG* & *IGF2*) from 24 h onwards (Fig. 4d). Thus, this landmark corresponds to cells that are effectively committed to becoming fully mature skeletal muscle cells. The final landmark consisted of cells exhibiting the lowest level of potency and emerged only at 48 h, becoming most prominent at 72 h (Fig. 4d). As with the previous landmark, cells in this group also expressed myogenic markers, and likely represent a terminally differentiated and more mature state of skeletal muscle cells. In summary, SCENT inferred lineage trajectories that are highly consistent with known biology and with those obtained by previous algorithms such as Monocle[14] and MPath[15]. However, in contrast to Monocle and MPath, SCENT inferred these reconstructions without the explicit need of knowing the time-point at which samples were collected.

**SCENT detects drug resistant cancer stem cell phenotypes.** Cancer cells are known to be less differentiated and to acquire a more plastic phenotype compared to non-malignant cells. Hence their signalling entropy should be higher than that of

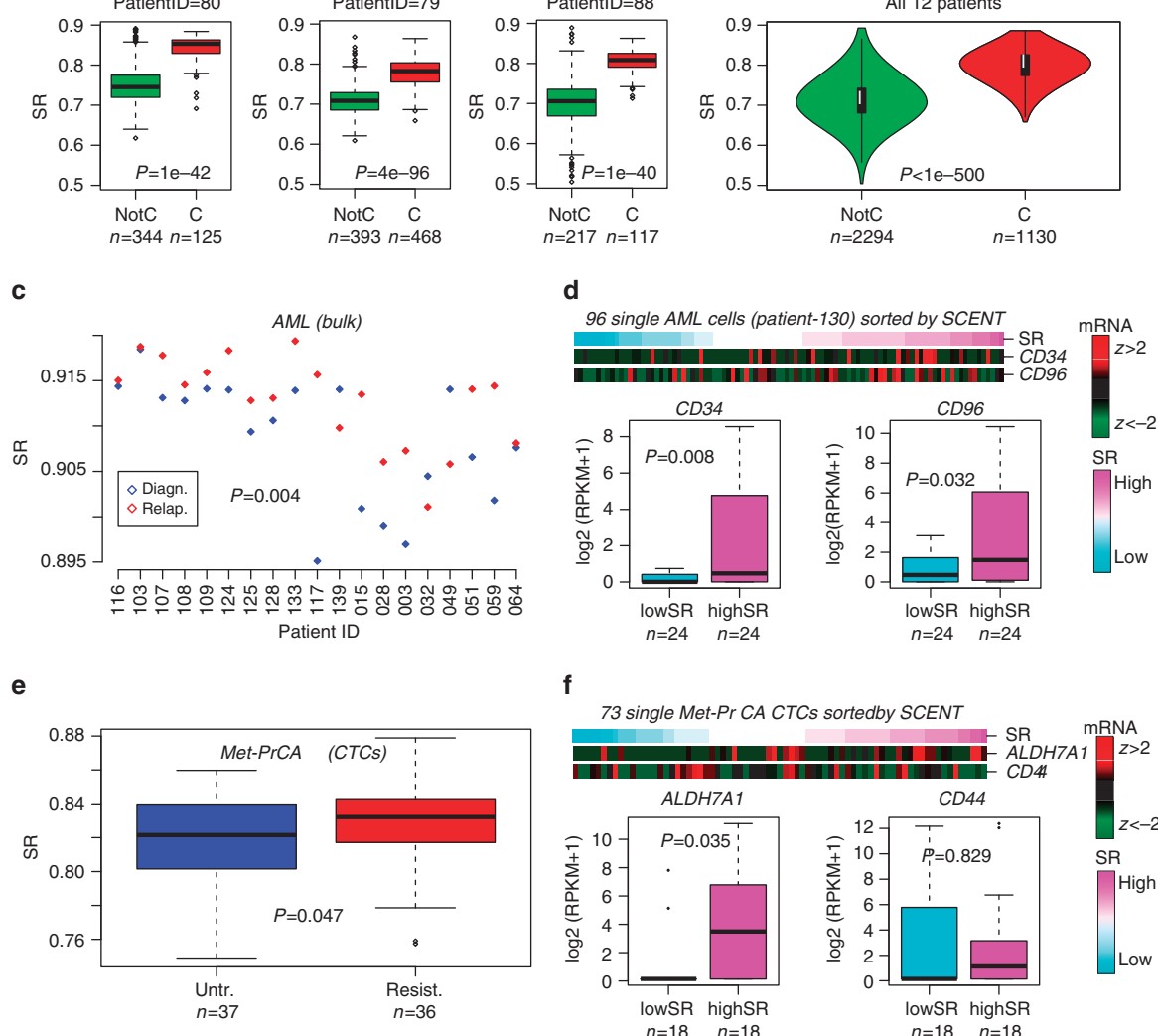

**Figure 5 | Increased signalling entropy in cancer cells and identification of drug resistant cancer stem cells.** (**a**) Boxplots of the signalling entropy (SR) for single melanoma cancer cells (**c**) compared to non-malignant (NotC) cells for three different melanoma patients (patient IDs given above each plot). Numbers of single cells are given below each boxplot. *P* value is from a Wilcoxon rank-sum test. (**b**) As **a**, but now pooled across all 12 patients. (**c**) Comparison of signalling entropy (SR) of 19 diagnostic acute myeloid leukaemia bulk samples to relapsed samples from the same patients. Wilcox rank sum test *P* value (one-tailed paired) is given. (**d**) Sorting of 96 single-AML cells from one patient according to signalling entropy and comparison of mRNA expression of AML CSC markers between low and high SR groups. *P* values from a one-tailed Wilcox test. (**e**) Comparison of signalling entropy (SR) of circulating tumour cells from metastatic prostate cancer patients who did not receive AR inhibitor treatment (UNTR) to those which developed resistance (RESIST). *P* value from a one-tailed Wilcox test. (**f**) Sorting of 73 single CTCs according to SCENT (signalling entropy, SR) into low and high SR groups. Correlation of gene expression of one putative CSC marker (ALDH7A1) with SR.

non-malignant cell-types. We confirmed this using scRNA-Seq data from 12 melanomas (Melanoma set[23], Supplementary Table 1), for which sufficient normal and cancer cells had been profiled (Fig. 5a, Supplementary Fig. 11). Although there was some variation in the signalling entropy of cancer cells between individuals, this variation was relatively small in comparison to the difference in entropy between cancer and normal cells. Combining data across all 12 patients, demonstrated a dramatic increase in the signalling entropy of single cancer cells compared to non-malignant ones (Wilcoxon rank-sum test $P < 1e - 500$, Fig. 5b).

Since signalling entropy is increased in cancer and correlates with stemness, it could, in principle, be used to identify putative cancer stem cells (CSC) or drug resistant cells. To test this, we first computed and compared signalling entropy values for 38 acute myeloid leukaemia (AML) bulk samples from 19 AML

patients, consisting of 19 diagnostic/relapse pairs[43]. Confirming that signalling entropy marks drug resistant cell populations, we observed a higher entropy in the relapsed samples (paired Wilcox test $P = 0.004$, Fig. 5c). For one relapsed sample, scRNA-Seq for 96 single-AML cells was available (AML set, Supplementary Table 1). We posited that comparing the signalling entropy values of these 96 cells would allow us to identify a CSC-like subset responsible for relapse. Since in AML there are well accepted CSC markers (*CD34, CD96*), we tested whether expression of these markers in high entropy AML single cells is higher than in low entropy AML single cells (Fig. 5d). Both *CD34* and *CD96* were more highly expressed in the high entropy AML single cells (Wilcox test $P = 0.008$ and 0.032, respectively, Fig. 5d).

We next computed signalling entropies for 73 CTCs derived from 11 castration resistant prostate cancer patients (CTC-PrCa set, Supplementary Table 1), of which five patients exhibited

progression under treatment with enzalutamide (an androgen receptor (AR) inhibitor) ($n = 36$ CTCs), with the other six patients not having received treatment ($n = 37$ CTCs)[44]. Although of marginal significance, signalling entropy was higher in the CTCs from patients exhibiting resistance (Wilcox test $P = 0.047$, Fig. 5e). Among putative prostate CSC markers (for example, *CD44, CD133, KLF4* and *ALDH7A1*)[44], we observed a positive association of signalling entropy with *ALDH7A1* expression, suggesting that *ADLH7A1* (and not other markers such as CD44) may mark specific prostate CSCs which are resistant to enzalutamide treatment (Fig. 5f).

**Regulation of single-cell expression heterogeneity.** It has been proposed that expression heterogeneity of cell populations is regulated in the sense that the transcriptomes of individual cells within the population differ in a manner which optimizes an objective function, such as pluripotency or homeostasis[3]. To test whether signalling entropy can predict such regulated expression heterogeneity, we compared the distribution of single-cell entropies to the signalling entropy of the bulk population. Specifically, we devised a 'measure of regulated heterogeneity' (MRH), which measures the likelihood that the signalling entropy of the cell population could have been observed from picking a single cell at random from that population (online Methods, Fig. 6a). We first estimated MRH for the data from Chu *et al.*, for which matched bulk and scRNA-Seq data is available. We first note that although for bulk samples entropy differences between cell-types were smaller, that they were nevertheless consistent with the trends seen at the single-cell level (Supplementary Fig. 12, Fig. 2c). The MRH for each of the six cell-types (hESCs, NPCs, DEPs, TBs, HFFs and ECs) in Chu *et al.*, revealed evidence of regulated heterogeneity, with the entropy values of bulk samples being significantly higher than that of single cells (Fig. 6b). As a negative control, the signalling entropy of the average expression over bulk samples did not exhibit regulated heterogeneity (normal deviation test $P = 0.30$, Fig. 6b), as required since bulk samples are not linked in space or time and represent non-interacting cell populations.

We note that for the previous analysis, matched bulk RNA-Seq data is not absolutely required since bulk samples can be approximated by averaging the expression profiles of individual cells in the population. We verified this, although, as expected, the entropy values for the true bulk samples were always marginally higher, in line with the fact that single-cell assays only capture a subpopulation of the bulk sample (Fig. 6c). We also verified that MRH results were not driven by the larger number of dropouts in scRNA-Seq data. Specifically, we simulated bulk samples by aggregating single cells representing the same cell-type and then resampling transcript counts matching to the average number of transcripts seen in single cells (Methods). We observed that signalling entropy of the simulated bulk did not alter appreciably upon downsampling and that results were unchanged (Supplementary Fig. 13).

Next, we repeated the MRH analysis for T-cells and B-cells found in melanomas (Melanoma set, Supplementary Table 1), for which sufficient numbers of single cells had been profiled. In all cases, signalling entropies of the bulk were much higher than expected based on the distribution of single-cell entropies (Supplementary Fig. 14). Evidence for regulated expression heterogeneity was also seen among the melanoma cancer cells from each of 12 patients (combined Fisher test $P < 1\mathrm{e} - 6$, Supplementary Fig. 15). We also analysed RNA-Seq data for 96 single cancer cells from a relapsed patient with acute myeloid leukaemia(AML set[43], Supplementary Table 1). The signalling entropy for the AML cell population was 0.88, significantly larger

than the maximal value over the 96 cells (SR = 0.82, Normal deviation test $P < 0.001$, Fig. 6d). Again, as a negative control we analysed all 19 bulk AML samples at relapse and diagnosis, treating bulk samples from independent AML patients as if they were single cells from a common population. Estimating the signalling entropy of the average expression profile over all 19 bulk samples did not reveal a value significantly higher than that of the individual bulk samples (normal deviation test $P = 0.32$, Fig. 6d).

## Discussion

Although Waddington proposed his famous epigenetic landscape of cellular differentiation many decades ago[10], it has proved challenging to construct a robust molecular correlate of a cell's elevation in this landscape. Here we have made significant progress, demonstrating that the differentiation potency and phenotypic plasticity of single cells, be they normal or malignant, can be estimated *in silico* from their RNA-Seq profile using signalling entropy. As we have seen, signalling entropy can accurately discriminate pluripotent from multipotent and differentiated cells, without the need for feature selection or training, outperforming a pluripotency gene expression signature and providing a more general measure of differentiation potency.

Importantly, signalling entropy should not be confused with other transcriptional entropy measures, which are estimated over populations of single cells[45,46]. For instance, the 'transcriptional entropy' of Richard *et al.*[45] is estimated for single genes across single cells, and therefore reflects the amount of intercellular heterogeneity in the expression of a given gene. Our signalling entropy measure is estimated for a single cell across genes in the context of a large gene network, which therefore incorporates systems-level information and is genome-wide (Fig. 1a,b). While the signalling entropy of single cells will influence the amount of transcriptional heterogeneity and entropy as defined by Richard *et al.*, the precise relation between the two entropies is non-trivial. Indeed, we have here shown how we can assign single cells into potency states, from which a SI over the whole cell population (that is, using the distribution of potency states over single cells) can then be estimated (Fig. 1c). This SI is more analogous to the transcriptional entropy of Richard *et al.* Indeed, we have shown how this SI is higher in a population of NPCs than in a population of hESCs (Fig. 3c). Thus, the SI has nothing to do with potency as such, that is, it does not measure the average differentiation potency of single cells in a cell population. In contrast, our signalling entropy does measure potency of single cells in a cell population. Thus, there is no requirement for our single-cell signalling entropy measure to exhibit a peak before a critical cell-fate transition occurs[45,46]. In contrast, the SI of a cell population derived from signalling entropy may exhibit the expected hallmarks of criticality. It will be interesting in future to test this with upcoming high-resolution time course and genome-wide scRNA-Seq data.

The ability of signalling entropy to independently order single cells according to differentiation potency is a central component of the SCENT algorithm, which, as shown here, can help quantify and identify biologically relevant intercellular expression heterogeneity and cell subpopulations. Indeed, key findings which strongly support the validity of SCENT are the following: (i) using SCENT we were able to correctly predict that a hESC population contains a small fraction of cells of lower potency which are primed for differentiation, (ii) SCENT inferred that an assayed NPC population was made up two distinct subsets, correctly predicting that only the lower potency subset represents *bona fide* NPCs (as determined by expression of known neural stem cell markers) and (iii) in a time-course

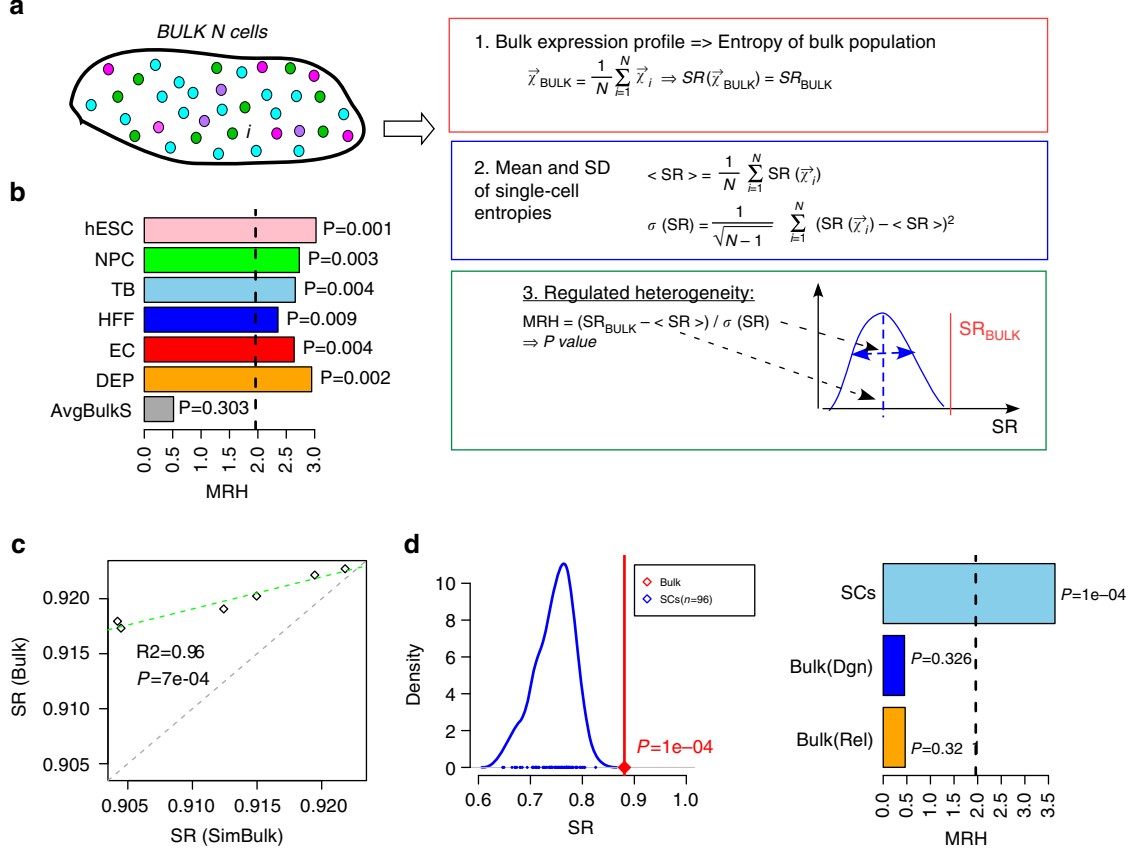

**Figure 6 | Signalling entropy predicts regulated expression heterogeneity of single-cell populations.** (**a**) Definition of the measure of regulated expression heterogeneity (MRH). The MRH is a z-statistic, obtained by measuring the deviation of the signalling entropy (SR) of the bulk expression profile from the mean of single-cell entropies, taking into account the variability of single-cell entropies in the population. (**b**) Barplots of MRH for each cell-type population from Chu et al., representing the degree to which the signalling entropy of the cell population is higher than that of single cells. P values are from a one-tailed normal-deviation test. Dashed line indicates the line $P = 0.05$. AvgBulkS compares the signalling entropy of the average expression over all bulk samples to that of the individual bulk samples, indicating that although the RHM is positive (signalling entropy increases), that it is not significantly higher than that of the individual bulk samples. (**c**) Scatterplot of the signalling entropy of bulk samples (y-axis), representing six cell-types (hESCs, NPCs, DEPs, TBs, HFFs, ECs) against the corresponding signalling entropies of these cell populations obtained by first averaging the expression profiles of single cells ('Simulated Bulk', x-axis). $R^2$ value and P value are given with green dashed line representing the fitted regression. Observe how the signalling entropy of bulk samples is always higher than that obtained from first averaging expression of single cells, in line with the fact that the assayed single cells are a subpopulation of the full bulk sample. (**d**) Left panel: comparison of the signalling entropy of an acute myeloid leukemia (AML) bulk sample (red line and point) to the signalling entropies of 96 single-AML cells (blue) from that bulk sample. P value is from a one-tailed normal deviation test. Right panel: comparison of the MRH value for the matched 96 single cells and bulk AML sample (SCs) to the MRH values obtained by comparing the signalling entropy of the average expression over 19 AML bulk samples to the signalling entropies of each individual AML bulk sample. The 19 AML bulk samples come in pairs, obtained at diagnosis (dgn) and relapse (rel), which are shown separately. P values are from a one-tailed normality deviation test.

differentiation experiment of human myoblasts, SCENT correctly identified a contaminating interstitial mesenchymal cell population, whose potency did not change appreciably during the experiment. We note that this particular insight is not readily obtainable using other algorithms such as Monocle or MPath[14,15]. Thus, the ability of SCENT to assign single cells and cell subpopulations to specific potency states thus adds novel insight and functionality over what can be achieved with other existing algorithms. Alternatively, signalling entropy could be combined with existing algorithms like Monocle[14] or DPT[17,47] to empower their inference, since signalling entropy provides a more unbiased, independent, approach to ordering single cells in pseudo-time, that is, it constitutes an approach which does not need prior knowledge such as the time point or markers of specific cell-types.

In a proof of principle analysis, we further demonstrated the ability of SCENT to identify putative drug resistant CSCs, encompassing two different cancer types (AML and prostate

cancer), including CTCs. The ability to quantify stemness in cancer cell populations, either in tissue or in circulation, is a task of enormous importance. As shown here, as well as in our previous work on bulk cancer tissue[9,11,13], signalling entropy is, so far, the only single sample measure to have been conclusively demonstrated to robustly correlate with stemness in both normal and cancer cells. Indeed, a recent study by Gruen et al.[18] explored a very different measure of transcriptome entropy, but which was not demonstrated to correlate well with differentiation potency or cancer. Likewise, signalling entropy is a more general measure of stemness/plasticity outperforming existing pluripotency expression signatures, as shown here and previously[11].

Importantly, signalling entropy also provides a computational framework in which to understand differentiation potency at the macroscopic (cell population) level from the corresponding potencies of single cells. As shown here, signalling entropy of cell populations, be they normal or malignant cells, exhibit synergy, with the entropy of the bulk being substantially higher

than the entropy values of single cells. While no existing assay can measure all single cells in a population, we nevertheless demonstrated that our result is non-trivial, since mixing up bulk samples (to serve as a negative control) did not reveal such synergy. We also showed that these results were not confounded by the larger number of dropouts in scRNA-Seq data. Biologically, increased potency of a cell population as a result of synergistic cell–cell interactions, supports the view that features such as pluripotency are best understood at the cellular population level[3].

Finally, it is important to discuss the technical and biological properties of signalling entropy that underlie its robustness as a measure of differentiation potency. First of all, gene expression values enter the computation of signalling entropy only as gene ratios. Taking ratios of gene expression values and introducing a regularization term to offset dropouts, makes the resulting inference much less sensitive to the sequencing depth, absolute scale and normalization procedure of scRNA-Seq data. Second, signalling entropy is estimated over a fairly large number of genes (8,000–10,000), making it naturally robust to single gene dropouts. Third, its biological robustness stems in part from differentiation potency being encoded by a subtle positive correlation between the transcriptome and connectome, similar to our previous observations in the context of cancer[12]. Since there is no reason to expect that technical dropouts in scRNA-Seq should correlate with the connectivity of the corresponding protein in a PPI network, such technical effects are expected to average out. Finally, it is worth emphasizing in this context that signalling entropy provided a more accurate and robust measure of differentiation potency than other transcriptomic entropy-based measures (those used in StemID and SLICE) which do not use network information.

To conclude, signalling entropy and the SCENT algorithm provide a computational framework to advance our understanding of single-cell biology. We envisage that SCENT will be of great value for quantifying biologically relevant intercellular heterogeneity and for identifying putative normal and cancer stem-cells from scRNA-Seq data.

## Methods

**Single cell and bulk RNA-Seq data sets.** The main data sets analysed here, the NGS platform used and their public accession numbers are listed in Supplementary Table 1. Below is a more detailed description of the samples in each data set:

*Chu et al. set.* This RNA-Seq data set derives from Chu et al.[28] This set consisted of four experiments. Experiment-1 generated scRNA-Seq data for 1,018 single cells, composed of 374 hESCs (212 single cells from H1 and 162 from H9 cell line), 173 NPCs, 138 DEPs, 105 mesoderm-derived ECs, 69 TB cells and 159 HFFs. Experiment-2 is a time course differentiation of single cells, specifically of hESCs induced to differentiate into the definite endoderm, via a mesoendoderm intermediate. Time points assayed were before induction ($t = 0$ h, $n = 92$), 12 h after induction (12 h, $n = 102$), 24 h ($n = 66$), 36 h ($n = 172$), 72 h ($n = 138$) and 96 h ($n = 188$). Experiment-3 matches experiment-1 and consists of RNA-Seq data from 19 bulk samples: 7 representing hESCs, 2 representing NPCs, 2 TBs, 3 HFFs, 3 ECs and 2 DEPs. Experiment-4 consists of 15 RNA-Seq profiles from bulk samples, profiled as part of the time-course differentiation experiment (Experiment-2), with three samples per time-point (12 h, 24 h, 36 h, 72 h, 96 h).

*Melanoma set.* This scRNA-Seq data set derives from Tirosh et al.[23], and consists of 4,645 single cells derived from the tumour microenvironment of 19 melanoma patients. Of these, 3,256 are non-malignant cells, encompassing T-cells ($n = 2,068$), B-cells ($n = 515$), NK cells ($n = 52$), Macrophages ($n = 126$), endothelial cells (EndC, $n = 65$) and CAFs ($n = 61$). The rest of single cells profiled were malignant melanoma cells ($n = 1,257$).

*AML set.* This set derives from Li et al.[43] A total of 96 single cells from a relapsed AML patient (patient ID $= 130$) were profiled. In addition, 38 paired bulk AML samples were profiled from 19 patients (all experiencing relapse), with 19 samples obtained at diagnosis and with the other matched 19 samples obtained at relapse.

*hESC set.* This set derives from Yan et al.[30] It consists of 124 single-cell profiles, of which 90 are from different stages of embryonic development, with 34 cells representing hESCs. These 34 hESCs were derived from the inner cell mass, with eight cells profiled at primary outgrowth and 26 profiled at passage-10. The 90

single cells from the pre-implantation embryo were distributed as follows: Oocyte ($n = 3$), Zygote ($n = 3$), 2-cell embryo ($n = 6$), 4-cell embryo ($n = 12$), 8-cell embryo ($n = 20$), morulae ($n = 16$) and late blastocyst ($n = 30$).

*Trapnell et al. set.* This scRNA-Seq set derives from Trapnell et al.[14] It consists of a time-course differentiation experiment of human myoblasts, which profiled a total of 372 single cells: 96 cells at $t = 0$ (time at which differentiation was induced), 96 at $t = 24$ h after induction, another 96 at $t = 48$ h after induction and 84 cells at 72 h post induction.

*CTC-PrCa set.* This scRNA-Seq data set derives from Miyamoto et al.[44] We focused on a subset of 73 single cells from castration resistant prostate cancers, of which 36 derived from patients who developed resistance to enzulatamide treatment, with the remaining 37 derived from treatment-naïve patients.

*Treutlein set.* This scRNA-Seq data set derives from Treutlein et al.[29] There are a total of 201 single cells assayed at four different stages in the developing mouse epithelium, including embryonic day 14, 16, 18 and adulthood. At E18, a subset of single cells were characterized into alveolar type-1 and type-2 cells (AT1 & AT2), as well as a putative bipotent (BP) subgroup.

**The single-cell entropy algorithm.** There are five steps to the SCENT algorithm: (1) estimation of the differentiation potency of single cells via computation of signalling entropy, (2) inference of the potency state distribution across the single-cell population, (3) quantification of the intercellular heterogeneity of potency states, (4) inference of single-cell landmarks, representing the major potency-coexpression clusters of single cells and (5) lineage trajectory (or dependency network) reconstruction between landmarks. We now describe each of these steps:

*Computation of signalling entropy.* The computation of signalling entropy for a given sample proceeds using the same prescription as used in our previous publications[9,11]. Briefly, the normalized genome-wide gene expression profile of a sample (this can be a single cell or a bulk sample) is used to assign weights to the edges of a highly curated PPI network. The construction of the PPI network itself is described in detail elsewhere[11], and is obtained by integrating various interaction databases which form part of Pathway Commons (www.pathwaycommons.org)[48]. The weighting of the network via the transcriptomic profile of the sample provides the biological context. The weight of an edge between protein $i$ and protein $j$, denoted by $w_{ij}$, is assumed to be proportional to the normalized expression levels of the coding genes in the sample, that is, we assume that $w_{ij} \sim x_i x_j$. We interpret these weights (if normalized) as interaction probabilities. The above construction of the weights is based on the assumption that in a sample with high expression of $i$ and $j$, that the two proteins are more likely to interact than in a sample with low expression of $i$ and/or $j$. Viewing the edges generally as signalling interactions, we can thus define a random walk on the network, assuming we normalize the weights so that the sum of outgoing weights of a given node $i$ is 1. This results in a stochastic matrix, $P$, over the network, with entries

$$p_{ij} = \frac{x_j}{\sum_{k \in N(i)} x_k} = \frac{x_j}{(Ax)_i},$$

where $N(i)$ denotes the neighbours of protein $i$, and where $A$ is the adjacency matrix of the PPI network ($A_{ij} = 1$ if $i$ and $j$ are connected, 0 otherwise, and with $A_{ii} = 0$). The signalling entropy is then defined as the entropy rate (denoted Sr) over the weighted network, that is,

$$\text{Sr}(\vec{x}) = -\sum_{i=1}^{n} \pi_i \sum_{j \in N(i)} p_{ij} \log p_{ij},$$

where $\pi$ is the invariant measure, satisfying $\pi P = \pi$ and the normalization constraint $\pi^T 1 = 1$. The invariant measure, also known as steady-state probability, represents the relative probability of finding the random walker at a given node in the network (under steady-state conditions that is, long after the walk is initiated). Nodes with high values thus represent nodes that are particularly influential in distributing signalling flux in the network. In the steady-state we can assume detailed balance (conservation of signalling flux, that is, $\pi_i p_{ij} = \pi_j p_{ji}$), and it can be shown[9] that $\pi_i = x_i(Ax)_i/(x^T Ax)$. Given a fixed adjacency matrix $A$ (that is, fixing the topology), it can also be shown[9] that the maximum possible Sr among all compatible stochastic matrices $P$, is the one with $P = \frac{1}{\gamma} v^{-1} \otimes A \otimes v$ where $\otimes$ denotes product of matrix entries and where $v$ is the dominant eigenvector of $A$, that is, $Av = \lambda v$ with $\lambda$ the largest eigenvalue of $A$. We denote this maximum entropy rate by maxSr, and define the normalized entropy rate (with range of values between 0 and 1) as

$$\text{SR}(\vec{x}) = \frac{\text{Sr}(\vec{x})}{\text{maxSr}}.$$

Throughout this work, we always display this normalized entropy rate.

*Inference of potency states.* In this work, we show that signalling entropy (that is, the entropy rate SR) provides a proxy to the differentiation potential of cells. We can model a cell population as a statistical mechanical model, in which each single cell has access to a number of different potency states. For a large collection of single cells we can estimate their signalling entropies, and infer from this distribution of signalling entropies the number of underlying potency states using a mixture modelling framework. Since SR is bounded between 0 and 1, we first conveniently transform the SR value of each single cell into their logit scale, that is,

$y(SR) = \log_2(SR/(1-SR))$. Subsequently, we fit a mixture of Gaussians to the $y(SR)$ values of the whole cell population, and use the Bayesian information criterion (as implemented in the *mclust* R-package)[49] to estimate the optimal number $K$ of potency states, as well as the state-membership probabilities of each individual cell. Thus, for each single cell, this results in its assignment to a specific potency state.

*Quantifying intercellular heterogeneity of potency states.* For a population of $N$ cells, we can then define a probability distribution $p_k$ over the inferred potency states. For $K$ inferred potency states, one can then define a normalized SI:

$$SI = -\frac{1}{\log K}\sum_{k=1}^{K} p_k \log p_k,$$

which measures the amount of heterogeneity in potency within the single-cell population ($1$ = high heterogeneity in potency, $0$ = no heterogeneity in potency).

*Inference of co-expression clusters and landmarks.* With each cell assigned to a potency state, we next perform clustering (using the scRNA-seq profiles) of the single cells. We use the partitioning-around-medoids (PAM) algorithm with the average silhouette width to estimate the optimal number of clusters, a combination which was found to be among the most optimal clustering algorithms in applications to omic data[50]. Clustering of the cells is performed over a filtered set of genes that are identified as those driving most variation in the complete data set, as assessed using singular value decomposition (SVD). In detail, we perform a SVD on the full $z$-scored normalized RNA-Seq profiles of the cells, selecting the significant components using random matrix theory (RMT)[51] and picking the top 5% genes with largest absolute weights in each significant component. The final set of genes is obtained by the union of those identified from each significant component. PAM clustering (with a Pearson distance correlation metric) of all cells results in the assignment of each cell into a co-expression cluster, with a total number of $n_p$ cell clusters. Thus, each cell is assigned to a unique potency state and co-expression cluster. Finally, landmarks are identified by selecting potency-state cluster combinations containing at least 1–5% of all single cells. Importantly, each of these landmarks has a specific potency state and mean signalling entropy value, allowing ordering of these landmarks according to potency.

*Inference of lineage trajectories.* For each landmark in step-4, we compute centroids of gene expression using only cells that are contained within that landmark and defined only over the genes used in the PAM clustering. Partial correlations[52,53] between the centroid landmarks are then estimated to infer trajectories/dependencies between landmarks. Significant positive partial correlations may indicate transitions between landmarks. Since each landmark has a signalling entropy value associated with it, directionality is inferred by comparing their respective potency states.

**A fast Pearson correlation approximation.** Under certain assumptions (to be discussed below), there is a useful approximation to signalling entropy, which also provides important biological insight. It entails first using an approximation for the steady-state probability (invariant measure) $\pi$. As before, in the steady-state, we can assume the detailed balance condition (conservation of signalling flux: that is, $\pi_i p_{ij} = \pi_j p_{ji}$), so that the invariant measure satisfies $\pi_i \sim x_i(Ax)_i$ (ref. 9). If we now take a global mean field approximation, that is, if we replace the expression values of the neighbours of gene $i$, with the mean expression value over all genes in the network, it then follows that $\pi_i \sim x_i k_i$, where $k_i$ is the connectivity of gene/protein $i$ in the network. Hence, $SR = \sum_i \pi_i S_i \sim \sum_i x_i k_i S_i$, which is effectively the three-way correlation between the transcriptome, connectome and local signalling entropies. If we assume further that the dynamic range of local signalling entropies $S_i = -\sum_{j \in N(i)} p_{ij} \log p_{ij}$ is small (which for realistic PPI networks is often the case[12]), and also assuming that the local entropies correlate positively with node-degree, we obtain that $SR \sim x_i k_i$, that is, the signalling entropy is approximately the Pearson correlation of the cells transcriptome and the connectome from the PPI network.

Importantly, we stress that (i) this approximation is an empirical one which works reasonably well for the realistic PPI networks considered here, and (ii) that the signalling entropy and its Pearson correlation approximation are not equivalent, since there exist networks where the two measures give widely different answers. In particular, if a network has scale-free topology, but with the hubs not connected to each other, then a positive correlation between expression and connectivity may not lead to a higher signalling entropy. For instance, if the low-degree nodes ('bottlenecks') linking the hubs have very low expression then signalling flux cannot be distributed over the network, leading to a lower entropy rate compared to an expression configuration where all genes have similar expression values (Supplementary Fig. 7). For realistic PPI networks, hubs are generally connected to each other and for these type of networks, the Pearson approximation works well. We note that for a 8,393 node network with 300,916 edges, the computation of SR for 100 samples takes $\sim 370\,s$ on an Intel Xeon CPU E3-1575M 3.00 GHz, whereas that of its Pearson correlation approximation only takes 1/10 s, thus although the approximation is computationally much faster, the computation of SR for 1 sample only takes about 4 s.

**Ranking genes according to differential local entropy.** Since signalling entropy is obtained as a weighted average over local signalling entropies (that is, $SR = \sum_i \pi_i S_i$) with the local entropies defined by $S_i = -\sum_{j \in N(i)} p_{ij} \log p_{ij}$, the latter can be used to identify genes in the network where the signalling flux

distribution differs between two phenotypes. Specifically, we use the normalized version of the local signalling entropy, defined by $NS_i = -\frac{1}{\log k_i}\sum_{j \in N(i)} p_{ij} \log p_{ij}$, which is bounded between 0 and 1, thus allowing genes of different connectivity to be compared. Thus, for each gene and each sample, we can compute a local entropy and genes can then be ranked according to the difference in local entropy using an empirical Bayes framework[11,54] to derive moderated $t$-statistics which reflect the significance in differential local entropy. Adjustment for multiple-testing was performed using the Benjamini–Hochberg procedure.

**Gene set enrichment analysis.** We performed GSEA on the top-ranked genes, ranked according to differential local entropy between pluripotent and non-pluripotent cells. Specifically, we focused on the genes exhibiting increased local signalling entropy in pluripotent cells, and focused on a range of thresholds (top 500, 600, 700, 800, 900 and 1,000) to assess robustness. Enrichment was performed using a one-tailed Fisher's exact test, as implemented by us previously[55]. Enrichment was assessed against the Molecular Signatures Database (http://software.broadinstitute.org/gsea/msigdb)[34].

**Application to mouse scRNA-Seq data.** In our application to mouse scRNA-Seq data, we first converted mouse gene Ensembl IDs into their human homologues using the AnnotationTools Bioconductor package[56]. Only those mapping to a unique human homologue were considered. The resulting set of genes were then integrated with our human PPI network.

**Estimation of cell-cycle and TPSC pluripotency scores.** To identify single cells in either the G1-S or G2-M phases of the cell-cycle we followed the procedure described in ref. 23. Briefly, genes whose expression is reflective of G1-S or G2-M phase were obtained from refs 32,33. A given normalized scRNA-Seq data matrix is then $z$-score normalized for all genes present in these signatures. Finally, a cycling score for each phase and each cell is obtained as the average $z$-scores over all genes present in each signature.

To obtain an independent estimate of pluripotency we used the pluripotency gene expression signature of Palmer et al.[22], which we have used extensively before[11]. This signature consists of 118 genes that are overexpressed and 39 genes that are underexpressed in pluripotent cells. The TPSC score for each cell with scRNA-Seq data is obtained as the $t$-statistic of the gene expression levels between the overexpressed and underexpressed gene categories. Optionally, the scRNA-Seq is $z$-score normalized beforehand and the $t$-statistic is obtained by comparing expression $z$-scores. However, we note that the $z$-score procedure uses information from all single cells, so the fairest comparison to signalling entropy means we ought to compare expression levels. We note that the TPSC scores obtained from $z$-scores or expression levels were highly correlated and did not affect any of the conclusions in this paper.

**Comparison analysis of bulk and single-cell RNA-Seq data.** Since SR can be computed for each single cell, one can compare the predicted entropies of bulk samples (cell population) to those of the single cells making up that population. To test whether the entropy of the bulk deviates markedly from that of single cells, we computed a $z$-score, by comparing the entropy of the bulk to that of the single cells where the latter distribution is modelled as a Gaussian. This $z$-score is called MRH, since it assesses whether the transcriptomes of single cells differ in a regulated synergistic manner, increasing entropy (potency) well above that of single cells. In the case where matched bulk samples were not available, we simulated bulk samples in two distinct ways. In one approach, we simply averaged the single-cell transcriptomes before computing SR. In a second approach, which corrects for the large number of dropouts present in scRNA-Seq data, by first aggregate the transcript counts of all single cells, and then downsample counts so as to match to the average number of transcripts per single-cell. Robustness to the specific downsampling draw was tested by performing 100 Monte-Carlo samplings.

**Other entropy measure proxies for differentiation potency.** Briefly, we describe two other entropy-based measures for approximating differentiation potency in a single-cell context, but which do not make use of a PPI network. One measure is part of the StemID algorithm[18]. However, the original StemID algorithm does not estimate differentiation potency of single cells. Instead it provides estimates for single-cell clusters, which are inferred by clustering the expression profiles of single cells. Thus, for a given cluster $k$, StemID computes a potency which is proportional to $\delta E_k$, where

$$\delta E_k \equiv \text{median}_{c \in k}(E_c) - \min_l(\text{median}_{c \in l}(E_c)),$$

where $E_c$ is the information entropy of cell $c$, defined by $E_c = -\sum_{g=1}^{N} q_{gc} \log q_{gc}$ (where $N$ is the number of genes and where $q_{gc}$ is the normalized number of reads mapping to gene $g$ in cell $c$). Thus, to objectively compare to our signalling entropy measure, which does not use information of other cells when estimating potency of a given cell, we here use $E_c$ as the potency estimate from StemID. Another information entropy-based measure is part of the SLICE algorithm, proposed by Guo et al.[31] Briefly, in this approach, genes are first clustered into related GO-terms to define $m$ functional gene clusters. For a given cell $c$, relative

activity of each functional cluster $k$ is estimated from the average expression of genes mapping to that cluster. These activity scores are then normalized so that they can be interpreted as probabilities $q_{kc}$, and subsequently the potency of cell $c$ is estimated as the information entropy $H_c = E_B[-\sum_{k=1}^{m} q_{kc} \log q_{kc}]$ where the expectation is taken over a number of bootstraps over genes. We compute this information entropy using the R-script provided in Guo et al.[31]

**Code availability.** Signalling entropy is available as part of the Single Cell Entropy (SCENT) R-package and is freely available from github: https://github.com/aet21/SCENT.

**Data availability.** All data analysed in this manuscript is already publicly available from the following GEO (www.ncbi.nlm.nih.gov/geo/) accession numbers: GSE72056, GSE83533, GSE75748, GSE36552, GSE52529, GSE67980 and GSE52583. All data is also available on request from the authors.

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

## Acknowledgements

This work was supported by NSFC (National Science Foundation of China) grants, grant numbers 31571359 and 31401120 by a Royal Society Newton Advanced Fellowship (NAF project number: 522438, NAF award number: 164914) and by a Medical Research Council grant (number 519159). The author also wishes to thank Guo-Cheng Yuan for stimulating discussions.

## Author contributions

Manuscript was conceived and written by A.E.T. Statistical analyses were performed by A.E.T. T.E. contributed useful feedback.

## Additional information

**Competing interests:** The authors declare no competing financial interests.

