## [Peer Review File · Nature Communications]

Reviewers' comments:

Reviewer #1 (Remarks to the Author):

This paper presents a novel way to assess the potency of individual cells using single cell RNA-Seq expression profiles. This problem is very relevant, and a good solution would be very useful. The main idea of the paper is that high signalling entropy, appropriately defined, characterizes pluripotent cells and may be used to stratify heterogeneous populations or provide a pseudotemporal ordering for populations of differentiating cells in a unbiased way. The informal rationale for the proposed measure is that cells which have not committed to any particular lineage will "keep all options open", and express important genes in a promiscuous way. The key observation is that this promiscuity is not apparent by simply by examining the variability of gene expression on its own, but only becomes clear once the structure regulatory interactions between genes/proteins is also taken into account. The bulk of the paper consists of applying this idea to a number of different previously published datasets, including time-series data.

I found the paper to be interesting, well written and potentially useful but overly long in my opinion. I understand that the author is demonstrating the range of the proposed method but the essential idea seems to be very simple, and the volume of data presented obscured this essential simplicity. I would prefer to see the method illustrated using one or two key examples, with the remaining data presented in the supporting information. Additionally, I found the method to be presented in an unduly complex way. For example, it is observed that "It can be shown that signaling entropy is, in effect, the correlation of a cell's transcriptomic profile with the connectivity profile of the corresponding proteins in the PPI network". I take this to mean that if the genes encoding for hub proteins in the PPI network are highly expressed then the cell will gain a high signalling entropy score. In this case although it sounds complex, the signalling entropy is just a weighted average of gene expression, where the weighting is taken with respect to the connectivity of the underlying network. In the methods it is noted that "In this work, although we never use this approximation, in practice this approximation is highly accurate and helps understand the biological features of signaling entropy". If this is the case then the method is much simpler than presented, and I think that making it sound complex obscures, rather than aids, understanding. If the entropy is important then I think that it should be better justified (what does entropy add beyond the approximation described above?). If I have misunderstood then: 1) does the weighted average described above perform as well? (if not, why not?) and 2) I think that the rationale and mathematics should be described with greater precision, in order to better justify the method (see also my minor comment below).

Minor points:

- * There was little comparison of the method with other (perhaps simpler) methods (such as the weighted average described above). I think that this should be included to better justify the method.
- * A connection to the Waddington landscape was made but the nature of this connection is unclear - I understand that signalling entropy is interpreted as the "elevation" but what precisely does this mean?
- * The model is not presented in a mathematically transparent way. If the mathematical details are needed (i.e. if notions of entropy are really essential to the method [see comment above]), then they should be presented with more clarity. I found the mathematical details hard to follow and I suspect others will too. For example, I did not see how the matrix $w_{\{gh\}}$, representing the projection of the single cell data on to the PPI was used (it is not mentioned again after it is defined); statements such as "In a mean field approximation it is clear that ... $x_i k_i$ " were not clear to me (for example, what is k_i here?)

Reviewer #2 (Remarks to the Author):

The present study by Andrew E. Teschendorff extends the concept of signalling entropy to single-cell RNA-seq data. In previous papers from his group, the author has shown that signalling entropy permits the identification of differences in pluripotency levels between cell populations. In this paper, signaling entropy is convincingly demonstrated to reveal distinct pluripotency states across single cell transcriptomes and to facilitate the derivation of directed differentiation trajectories. It represents an interesting alternative use of the entropy concept to the derivation of cellular differentiation trajectories compared to other recently published methods (Grün et al., 2016, Cell Stem Cell; Guo et al., 2016, Nucleic Acids Research). The algorithm will be a useful tool for the community, enabling the de novo identification of stem cells from single cell transcriptome data.

The study is carefully done and technically sound; the manuscript is well structured and clearly written.

However, I have a few concerns that have to be addressed before I can recommend the manuscript for publication.

Major points:

1. The signaling entropy was shown to correlate with pluripotency levels of cell states arising from in vitro differentiation assays (hESCs, myoblasts) or cancer. A very important application would be the identification of adult tissue stem cells, which are in many cases still unknown. There are published datasets available that would allow testing the performance of the method for this type of stem cells, e. g. lung epithelial cells (Treutlein et al., 2014, Nature), glial cells (Pollen et al., 2015, Cell), or intestinal cells and hematopoietic cells (Grün et al., 2016, Cell Stem Cell). Application to the intestinal data set would be of particular interest, since this is a multi-lineage differentiation system. Moreover, the data were generated with unique molecular identifiers (UMI) and I am not sure whether this approach was used in any of the other tested datasets. UMIs have been shown to allow improved quantification of single cell transcriptomes due to the removal of amplification bias, and it is important to see that signalling entropy correlates equally well with pluripotency, if this technique is used.
2. Regarding the cell cycle analysis it is not clear to me, whether cells of the same type will be assigned to distinct pluripotency states depending on the cell cycle phase. A strong non-linear dependence of the signalling entropy on the cell-cycle phase was shown in Fig. S5. Although this observation is perhaps explained by an up-regulation of diverse signalling pathways at G1-S and G2-M, one should consider correcting for this behaviour, since the actual level of pluripotency should not depend on the cell cycle phase.
3. The author claims the existence of regulated heterogeneity by arguing, that the signalling entropy measured for bulk samples, or after aggregating single cell transcriptomes to construct a pseudo-bulk sample, significantly exceeds the signalling entropy sampled from single cells. It is unclear, how strongly this result is influenced by dropout events, which are known to be prevalent in single-cell transcriptomes. The strength of a link in the protein-protein network for a single cell could be diluted due to a dropout of a gene, and this is less likely to happen if single-cell samples are pooled, or bulk samples are analysed. In order to test this, the same comparison should be done using a dataset generated with unique molecular identifiers, e. g. the intestinal data from Grün et al., 2016, Cell Stem Cell, since this approach allows counting of transcripts. The effect of dropout events could be alleviated in this case, if the number of transcripts in the pseudo-bulk sample is down-sampled to the average number of transcripts in a single cell.
4. To facilitate the application of the algorithm, the author should provide up-to-date PPI matrices

at least for human and mouse in the github repository together with the R code.

Minor points:

1. It is unclear to me, which data are used to generate Figure 2G. An AUC of 1 suggests a complete separation of the SR distributions for progenitors and differentiated cells. However, the violin-plot in Figure 2F shows overlapping distributions of progenitors and differentiated cells.
2. I was surprised that a large number of NPCs in the pluripotent state expresses levels of HES1 equally high as NPCs of the non-pluripotent state (Figure 1G). How can this be explained?

Reviewer #3 (Remarks to the Author):

In this manuscript the author proposes to use signaling entropy in order to define a measure of a cell's pluripotency. To compute this measure the author combines information on protein-protein interactions (defining in this framework gene-gene connectivity) with single-cell gene expression data, which, compared to other methods in the field, does not require prior feature selection or clustering. This measure of pluripotency is then used to order cells according to their progression on a differentiation trajectory, and ultimately reconstruct lineage trajectories.

Overall, this work is scientifically sound and well presented. The method proposed in this paper might contribute to the interpretation of single-cell experiments. Nevertheless, the availability of competing methods like monocle or DPT, and recent papers published on transcriptional fluctuation near lineage bifurcation points, might limit the impact of this manuscript. I would support publication in Nature Communications if three major issues are resolved:

- While this approach, in particular the combination of the connectome with the transcriptome, is interesting I have a conceptual issue with this approach, which needs to be resolved before publication in Nature Communications. It relies on the idea of a monotonically decreasing entropy along the differentiation trajectory. Recently, two papers were published stating that differentiation decisions on the transcriptional level resemble "critical" transitions: A Richard et al., Plos Biol., (2016) and M Mojtahedi et al., Plos. Biol. (2016). While I am not entirely convinced that this conclusion follows rigorously from their observations, one of these papers (Richard et al.) reported an increase in transcriptional entropy shortly before the lineage decision is made, i.e. non monotonicity of transcriptional entropy. This is, at least superficially, in contradiction with the results reported here. While this might potentially be due to different definitions of the entropy, to avoid confusion in the field this discrepancy needs to be resolved or discussed in the manuscript.
- Along the same lines, I am not convinced that, in principle, single-cell entropy should monotonically decrease with decreasing potency. One might equally well think that pluripotency or multipotency are tightly regulated attractor states, such that entropy increases only intermittently before a fate decision. I also found confusing the use of the word entropy in relation with the shallowness of Waddington's epigenetic landscape (Figure 1A). If the potential wells in this level of description are more shallow (have a larger second derivative) this would correspond to a higher plasticity (or susceptibility, in the terminology of statistical mechanics) but not necessarily to a higher entropy. Here, the terminology needs to be clarified in order to avoid confusion.
- Further to a comparison to the measures reported in the mentioned publications it would strengthen the impact of this work if the presented algorithm was compared to some of the methods that are already widely used in the field (like monocle or DPT). In how far does this method present an advantage in terms of (for example) accuracy, identification of branching points or scalability to established methods?

Minor points:

- At several places the author reports very small p-values (e.g. $p < 1e-500$). Given the limited machine precision and the assumptions underlying the hypothesis testing I am skeptical that such

small p-values can be estimated with certainty. Please clarify.

- Line 88: the term "connectivity profile" is unclear in the context of a correlation. A more explicit definition of the signalling entropy in the main text could help avoid confusion.

- Line 227: I don't understand why Fig. 4a implies a stepwise reduction in signaling entropy. Couldn't the signaling entropy decrease smoothly between 0h and 24h?

- In general it would be helpful if terms like "transcriptomic heterogeneity" and "regulated heterogeneity" would be more precisely in the main text.

- Line 485: I could not find the definition of k_i

Response to Reviewers' comments:

Reviewer #1 (Remarks to the Author):

General Comment: I found the paper to be interesting, well written and potentially useful but overly long in my opinion. I understand that the author is demonstrating the range of the proposed method but the essential idea seems to be very simple, and the volume of data presented obscured this essential simplicity. I would prefer to see the method illustrated using one or two key examples, with the remaining data presented in the supporting information.

Response: We thank the reviewer for taking time to evaluate our manuscript. We appreciate the reviewer's point, but demonstrating wide applicability of the Signaling Entropy measure is in our view one of the key novel and appealing aspects of this manuscript. Indeed, to the best of our knowledge, no other study has proposed a single-cell in-silico approximation to differentiation potency which works for both normal and cancer cells. Hence, to showcase the usefulness of Signaling Entropy in identifying putative cancer stem cell phenotypes from CTC data is a highly unique feature of our manuscript which makes it of interest to a broader readership. Likewise, the comparison of signaling entropy of single cells to that of bulk samples is important for the consistency of the whole approach and sheds further novel insights. In view of this, although we appreciate the reviewer's point, we believe that this is ultimately a purely editorial decision. Given the utmost importance of having measures which may help to identify putative cancer stem cells, we think that it is in the interest of the scientific community at large to keep the cancer-associated data as a main figure in the manuscript.

General Comment: Additionally, I found the method to be presented in an unduly complex way. For example, it is observed that "It can be shown that signaling entropy is, in effect, the correlation of a cell's transcriptomic profile with the connectivity profile of the corresponding proteins in the PPI network". I take this to mean that if the genes encoding for hub proteins in the PPI network are highly expressed then the cell will gain a high signalling entropy score. In this case although it sounds complex, the signalling entropy is just a weighted average of gene expression, where the weighting is taken with respect to the connectivity of the underlying network. In the methods it is noted that "In this work, although we never use this approximation, in practice this approximation is highly accurate and helps understand the biological features of signaling entropy". If this is the case then the method is much simpler than presented, and I think that making it sound complex obscures, rather than aids, understanding. If the entropy is important then I think that it should be better justified (what does entropy add beyond the approximation described above?). If I have misunderstood then: 1) does the weighted average described above perform as well? (if not, why not?) and 2) I think that the rationale and mathematics should be described with greater precision, in order to better justify the method (see also my minor comment below).

Response: We understand the reviewer's point. There are at least 5 good reasons why it is preferable to use Signaling Entropy over its Pearson Correlation approximation: (1) First, signaling entropy is built on a rigorous mathematical and physical framework. Indeed, signaling entropy is defined in terms of the entropy rate of a weighted network, where the "entropy rate" has long been used in the field of complexity and network physics to understand global properties of diffusion processes on networks (see e.g. Gomez-Gardenes J et al "Entropy rate of diffusion processes on complex networks, Phys Rev E

2008). In our application, where the weights derive from the gene expression profile of the sample, one can think of signaling entropy as a global measure of the level of signaling interaction uncertainty or promiscuity. We note that the Pearson correlation approximation does not offer such a direct interpretation. (2) Second, it is important to realize that Signaling Entropy and the Pearson Correlation approximation are, mathematically speaking, not equivalent, i.e. there could be weighted networks where both measures give different answers. The Pearson correlation of the connectome and transcriptome is only an approximation which works extremely well on the datasets analysed here, however, there could be future datasets where the approximation does not hold as well. (3) Third, in all our previous papers where we analysed bulk-tissue samples (see Banerji CR et al Sci Rep 2013 & PLoS Comp Bio 2015, Teschendorff et al Methods 2014 & Sci Rep 2015) we have used the signaling entropy rate measure to approximate differentiation potency. Hence, it would be odd (and also wrong) to ditch signaling entropy for an approximation of it. (4) Fourth, while the Pearson correlation approximation is “simple”, this simplicity can also limit the applicability of the method: for instance, in Banerji et al Sci Rep 2013, we used detailed information of the underlying network (ie local signaling entropies) to identify signaling pathways which are implicated in differentiation. This information would not have been retrieved had we used the Pearson correlation approximation. (5) Finally, let us not forget that cells are complex systems, and differentiation potency is likely to be an “emergent” property. It would therefore be premature and wrong to conclude that the good performance of signaling entropy is entirely due to the correlation property. We agree that addressing this question is important but this will require testing in many more datasets and will require special data which is currently not available, and hence is well beyond the scope and purpose of this manuscript.

In response to the reviewer’s point, we now mention the Pearson Correlation approximation only as a means of providing a fast proxy to signaling entropy. Correspondingly, we have also tried to improve the clarity and rationale for using signaling entropy to approximate differentiation potential in Results and Fig.1.

Minor points:

** There was little comparison of the method with other (perhaps simpler) methods (such as the weighted average described above). I think that this should be included to better justify the method.*

Response: We don’t think that the comparison to the Pearson correlation approximation is that relevant for the reasons given earlier. However, we have decided to compare Signaling Entropy to two other proposed potency measures which were used in the StemID and SLICE algorithm. This new data is shown in a new display item (Table-1), which we reproduce below. This table demonstrates that Signaling Entropy outperforms the measures considered in the StemID and SLICE algorithms, in terms of the accuracy of discriminating more potent cells from less potent ones:

Dataset		Signaling Entropy	SLICE	StemID
scRNA-Seq				
Chu1 (PI > NonPI)	P	3e-132	~1	3e-58
	AUC	0.96	<0.5	0.79

Chu2 (0h > 96h)	P	2e-38	0.94	1e-22
	AUC	0.97	<0.5	0.86
Trapnell (0h>72h)	P	6e-9	0.0003	2e-10
	AUC	0.74	0.65	0.75
Treutlein (E14>Adult)	P	5e-27	6e-26	5e-27
	AUC	1	0.998	1
Bulk RNA-Seq				
Chu3 (PI > NonPI)	P	4e-5	0.001	0.76
	AUC	0.99	0.90	<0.5

Table legend: Comparison of Signaling Entropy to SLICE and StemID as measures of differentiation potency in scRNA-Seq and bulk RNA-Seq datasets. Table lists one-tailed Wilcoxon rank sum test P-values and associated (one-tailed) AUCs, testing whether entropy is higher in the pluripotent or multipotent cells compared to the less potent cells in various scRNA-Seq and bulk RNA-Seq datasets. In Chu1, the comparison is between pluripotent (hESCs, n=374, PI) and non-pluripotent (n=644, NonPI) single cells. In Chu2, the comparison is between hESCs (0h, n=92) and definite endoderm progenitors sampled 96h later (n=188). In Trapnell, the comparison is between human myoblasts (0h, n=96) and differentiated skeletal muscle cells (72h, n=84). In Treutlein, the comparison is between early lung progenitors (E14, n=45) and mature alveolar cells (n=46). In Chu3, the comparison is between bulk hESCs (n=7) and non-pluripotent samples (n=12).

* A connection to the Waddington landscape was made but the nature of this connection is unclear - I understand that signalling entropy is interpreted as the "elevation" but what precisely does this mean?

Response: We are interpreting "elevation" in Waddington's landscape as a measure of the overall differentiation potency of a cell, i.e. the number of cell-fates a cell can in principle convert to. Hence, hESCs would occupy the highest elevation, whereas terminally differentiated cells would occupy the lowest positions. Importantly, when comparing signaling entropies we ought to compare cells within a lineage, so e.g. hESCs -> Hematopoietic Stem Cells -> Common Lymphoid Progenitors -> Lymphocytes. We are not suggesting that Signaling Entropy may be that useful for comparing cells that all occupy the same differentiation stage, e.g. comparing terminally differentiated cells such B-cells and T-cells to each other, although this is still an open question. We acknowledge that panels Fig.1A & 1B were not optimal and that the legend to Fig.1A was somewhat terse, and so in response to the reviewer's point we have now included a few more details into Fig.1A & 1B, and have extended the legend accordingly.

* The model is not presented in a mathematically transparent way. If the mathematical details are needed (i.e. if notions of entropy are really essential to the method [see comment above]), then they should be presented with more clarity. I found the mathematical details hard to follow and I suspect others will too. For example, I did not see how the matrix $w_{\{gh\}}$, representing the projection of the single cell data on to the PPI was used (it is not mentioned again after it is defined); statements such as "In a mean field approximation it is clear that ... $x_i k_i$ " were not clear to me (for example, what is k_i here?)

Response: We apologize for this. We think that the confusion is caused by us switching notation, as earlier we use g and h to denote the genes/proteins, and later we switch to using i and j . To clarify here, $w_{\{gh\}}$, the weight of the edge connecting genes g and h , is proportional to the gene expression levels x_g and x_h of genes g and h , respectively, i.e. $w_{\{gh\}} \sim x_g x_h$. However, in order to turn $w_{\{hg\}}$ into a stochastic matrix (i.e. probabilities $p_{\{gh\}}$) we require that $\sum_h \{w_{\{gh\}}\} = 1$, which means that we need to normalize the weights by $\sum_h \{x_g x_h\} = x_g \sum_h \{x_h\}$. Thus, x_g cancels out as it appears in the denominator and numerator, and so we end up with $p_{\{gh\}} = x_h / \sum_k \{w_{\{gk\}}\}$. In the revised version, in order to avoid confusion, we have now expressed everything in terms of i and j .

We also acknowledge that understanding the mean field approximation formula is not obvious, as it is an empirical approximation. It is mainly justified by our previous work (Teschendorff et al Sci Rep 2015), where we showed that the signaling entropy rate is determined mainly by the invariant measure i.e. the steady state probability π , where $\pi P = \pi$ (with P being the stochastic matrix). We have now improved the clarity of this.

Reviewer #2 (Remarks to the Author)

General Comment: The algorithm will be a useful tool for the community, enabling the de novo identification of stem cells from single cell transcriptome data. The study is carefully done and technically sound; the manuscript is well structured and clearly written.

Response: We thank the reviewer for taking time to evaluate our manuscript and for recognizing the importance of our work.

Major points:

Comment 1. The signaling entropy was shown to correlate with pluripotency levels of cell states arising from in vitro differentiation assays (hESCs, myoblasts) or cancer. A very important application would be the identification of adult tissue stem cells, which are in many cases still unknown. There are published datasets available that would allow testing the performance of the method for this type of stem cells, e. g. lung epithelial cells (Treutlein et al., 2014, Nature), glial cells (Pollen et al., 2015, Cell), or intestinal cells and hematopoietic cells (Grün et al., 2016, Cell Stem Cell). Application to the intestinal data set would be of particular interest, since this is a multi-lineage differentiation system. Moreover, the data were generated with unique molecular identifiers (UMI) and I am not sure whether this approach was used in any of the other tested datasets. UMIs have been shown to allow improved quantification of single cell transcriptomes due to the removal of amplification bias, and it is important to see that signalling entropy correlates equally well with pluripotency, if this technique is used.

Response: We completely agree with the reviewer that the most powerful application of this method would be to the identification of stem-cell phenotypes where these are still unknown. However, this is also a request for new experimental data, and although we are currently in collaboration with various experimental groups, we have decided that the computational method presented here is too important and that it merits a separate paper. Indeed, demonstrating the value of Signaling Entropy can be done on existing scRNA-Seq data and we show that it has the required resolution. For instance, we show how

it (i) can identify a small number of cells within a hESC population which are not pluripotent, and which express a number of mesoderm and neural differentiation markers, suggesting that these cells are already primed for differentiation into these lineages (see SuppFig.S7 in our manuscript), and (ii) how it can identify distinct NPC subgroups of varying potency (as shown in SuppFig.S8). We also dedicate one main figure (Fig.5) to demonstrating that it can identify cancer stem-cell phenotypes in both bulk and scRNA-Seq data. Nevertheless, in response to the reviewer’s point, we have now also analysed the data from Treutlein et al (which profiled over 200 single cells), data of which is shown in a new panel Fig.2H and in a new SuppFig.S3. For convenience, we depict all this new data below, which confirms that Signaling Entropy (SR) decreases with increasing developmental stage in the developing mouse lung epithelium. The data further supports the view that SR has the resolution to detect a novel bipotent subgroup of alveolar type cells, although cell numbers are small and it is unclear whether the subtypes defined by Treutlein are “optimal” or that they define a “ground truth”. Importantly, the data in the left panel demonstrates that SR works in mice.

Figure legend: Left panel: Signaling entropy (SR, y-axis) as a function of developmental stage in the differentiation of the distal mouse lung epithelium. Number of single cells measured at each stage is given. Wilcoxon rank sum test P-values between embryonic day 14 (E14) and all other stages are given. Right panel: Signaling entropy (SR, y-axis) of alveolar type 1 and 2 cells as well as putative bi-potent progenitors (BP) at E18. Wilcoxon rank sum test P-values between AT1 and BP, and between AT2 and BP are given.

Unfortunately, the intestinal data that the reviewer recommends does not seem to be complete, as according to the two GEO accession numbers for this data (from Gruen D et al Cell Stem Cell 2016 and Nature 2015, i.e. GSE76408 and GSE62270) there are only 13 and 35 samples, respectively. On inspection, most of these are termed “replicates” and some are not even single-cell, so the total number of independent samples is rather small. We are puzzled by this, because in e.g. Fig.1D of Gruen D et al Cell Stem Cell 2016 there appear to be at least over 100 single cells, yet this data does not seem to be publicly available. In any case, although we acknowledge that amplification bias is a concern, this is not unique to single-cell RNA-Seq, but is also a concern for bulk RNA-Seq data. Furthermore, we don’t think that amplification bias has seriously undermined the results presented in landmark single cell studies such as those from Tirosh et al Nature & Science 2016 or Patel A et al Science 2014 (data which we reanalyze here), that it discredits our results. A much more serious related concern, which the reviewer

mentions below, is the large number of dropouts in scRNA-Seq data, and we have addressed that concern as explained in detail further below.

Comment 2. Regarding the cell cycle analysis it is not clear to me, whether cells of the same type will be assigned to distinct pluripotency states depending on the cell cycle phase. A strong non-linear dependence of the signalling entropy on the cell-cycle phase was shown in Fig. S5. Although this observation is perhaps explained by an up-regulation of diverse signalling pathways at G1-S and G2-M, one should consider correcting for this behaviour, since the actual level of pluripotency should not depend on the cell cycle phase.

Response: The reviewer has raised an excellent question. In our experience, the cell-cycle G1-S and G2-M scores derived from gene expression data generally correlate very strongly with differentiation potency. For instance, we observe this to be the case in the Chu et al data where hESCs and multipotent progenitors representing the mesoderm, endoderm and ectoderm were profiled: so, for example, 371 out of 374 single hESCs exhibited a G1-S score larger than 0, whereas only 5 of 105 endothelial cells (mesoderm progenitors) did so. For this reason, we think that the more relevant question is whether we can make sense of the potency subgroups inferred using signaling entropy within single cell populations representing the same cell-type. And this question we already addressed in the manuscript as part of e.g. Fig.3G and SuppFigs.S7 &S8, which demonstrate that the subset of cells that exhibit lower entropies express higher levels of known differentiation markers.

Nevertheless, in response to the reviewer's point, we have performed three additional analyses in the Chu et al set to confirm that signaling entropy correlates with differentiation potency independently of cell-cycle scores. In one analysis, we have correlated signaling entropy (SR) to potency state (as determined by cell-type i.e. pluripotent hESCs vs all non-pluripotent cell types), adjusting for G1-S and G2-M cell-cycle scores. The results of this analysis are shown as a new SuppTable.S2, which demonstrates that signaling entropy still correlates with potency independently of cell-cycle scores. In our second analysis, we focused on all cells which exhibit a cell-cycle score less than 0.25, a threshold chosen to select low-cycling cells and to ensure reasonable numbers for each cell-type. Confirming the first analysis, this also supports the view that SR correlates with potency independently of cell-cycle phase. Finally, for each cell-type we also selected the 5% of cells with lowest cycling scores, and compared SR between pluripotent and non-pluripotent cell types, lending further support to our conclusions. The results of these last two analyses are shown in SuppTable.S2 as well as in new SuppFig.S6, which for convenience we reproduce below:

Figure legend: A) Signaling Entropy (SR) vs cell-type (pluripotent hESCs vs non-pluripotent cells), using only low-cycling cells as determined using a 0.25 threshold for the cell-cycle scores G1-S and G2-M. P-value is from a one-tailed Wilcoxon rank sum test, and the associated AUC classification accuracy statistic is given. **B)** As A), but for low-cycling cells defined as the cells with the lowest 5% cell cycle scores in each cell-type.

Thus, the high classification accuracy of pluripotent vs non-pluripotent cell-types is not lost by focusing only on those single-cells which are non-cycling.

Comment 3. The author claims the existence of regulated heterogeneity by arguing, that the signalling entropy measured for bulk samples, or after aggregating single cell transcriptomes to construct a pseudo-bulk sample, significantly exceeds the signalling entropy sampled from single cells. It is unclear, how strongly this result is influenced by dropout events, which are known to be prevalent in single-cell transcriptomes. The strength of a link in the protein-protein network for a single cell could be diluted due to a dropout of a gene, and this is less likely to happen if single-cell samples are pooled, or bulk samples are analysed. In order to test this, the same comparison should be done using a dataset generated with unique molecular identifiers, e. g. the intestinal data from Grün et al., 2016, Cell Stem Cell, since this approach allows counting of transcripts. The effect of dropout events could be alleviated in this case, if the number of transcripts in the pseudo-bulk sample is down-sampled to the average number of transcripts in a single cell.

Response: The reviewer has raised another excellent point, and we totally agree that the data and conclusions drawn from Fig.6 may seem premature. In our opinion there are two related but also separate issues here: (i) large dropout events in scRNA-Seq data, and (ii) amplification biases. Although we acknowledge that amplification bias may be a concern, this is a concern that applies to most of the published literature (i.e. the works of Tirosh I et al Nature & Science 2016, Patel AP Science 2014), and is not a concern which is unique to single-cell data. Hence, for the comparison of bulk to scRNA-Seq data, we don't feel that the second issue is such a major concern that it requires analysis of specific datasets. However, we totally agree that the first issue of large dropouts is a major concern and we agree that this could have driven the results shown in Fig.6. Hence, in order to address this particular concern, we have performed two additional analyses. First, for pseudo-bulk (or simulated bulk) samples we have down-sampled to the average number of transcripts in a single cell as suggested by the reviewer, and have redone the analysis shown in Fig.6B-C. The results of this analysis are shown in a new SuppFig.S12, which for convenience we reproduce below:

Figure Legend: **A)** Scatterplot of the signaling entropies (SR) for the bulk samples (“Bulk”) of Chu et al against the corresponding entropies estimated from simulated bulk samples which use scRNA-Seq data. The case where single-cell RNA-Seq data was averaged before computing SR is denoted by “SimBulk”, whereas the case where reads from the simulated bulk sample were downsampled to the average number of transcripts per single cell is denoted by “SimBulk-DownSampled”. R^2 values and P-values from a linear regression are given. There are 6 data points, one for each cell-type (hESC, NPC, HFF, TB, DEP, EC). **B)** For each cell-type this compares the distribution of single-cell signaling entropies (SR) (blue) to that of the simulated bulk sample (red line) using downsampling to correct for the larger number of dropouts in single-cell data. The z-statistic (MRH) and associated P-value are given.

This analysis therefore demonstrates that the large number of dropouts in scRNA-Seq data does not appreciably affect the SR values. To confirm this further, we have also tested the robustness of SR to the sequencing depth of single-cell data, observing that it is highly robust, even if only 20% of reads are

retained. For convenience, we show this data below,

Robustness of Signaling Entropy (SE) to resampling reads at reduced coverage. Scatterplots of SE at full coverage (Chu1 scRNA-Seq dataset, $n=1018$ single cells) against SE at reduced coverage, for 3 different sampling proportions (80%, 20% and 10% of full coverage) as indicated. Pearson Correlation Coefficient (PCC) is given.

We can understand the remarkable robustness of signaling entropy as follows: First, gene expression values enter the computation of SR as ratios, i.e. the stochastic matrix is defined by

$$p_{ij} = \frac{x_j}{\sum_{k \in N(i)} x_k} = \frac{x_j}{(Ax)_i}$$

which means that it is fairly insensitive to the absolute scale of scRNA-Seq data. We note that 0s from dropouts are always offset by a small integer (1) to avoid dividing by zero. With this regularization, the ratios of gene expression are therefore much more robust to sequencing depth. Second, SR is typically computed across 8000-10000 genes, which renders it less sensitive to dropouts affecting single genes. Moreover, there is no reason to expect a correlation between the dropout rate of a gene and its connectivity in a PPI network. In response to the reviewer's excellent point, we now emphasize in Discussion the robustness of SR and explain that its sole dependence on gene expression ratios is key to this robustness.

Comment 4. To facilitate the application of the algorithm, the author should provide up-to-date PPI matrices at least for human and mouse in the github repository together with the R code.

Response: In response to this, we now make two versions of our human PPI available on the github repository. These versions differ mainly in terms of the date when data was downloaded from PathwayCommons. As we explain in a separate manuscript, where we present Treutlein's data and show the validity of the method on mouse scRNA-Seq data, our approach is to map mouse genes to human homologs which then allows us to use the same human PPI.

Minor points:

1. *It is unclear to me, which data are used to generate Figure 2G. An AUC of 1 suggests a complete separation of the SR distributions for progenitors and differentiated cells. However, the violin-plot in Figure 2F shows overlapping distributions of progenitors and differentiated cells.*

Response: The ROC curves shown in old panel 2G (now new panel 2F) corresponds to the data shown in panel 2E, and the reviewer is right that there is some little overlap and that the AUC is not strictly speaking “1”, but 0.998. The reason why it showed as “1” is because we tend to round AUC values to two significant digits after the decimal point. In response to the reviewer, we now show the exact AUC value which is 0.998.

2. *I was surprised that a large number of NPCs in the pluripotent state expresses levels of HES1 equally high as NPCs of the non-pluripotent state (Figure 3G). How can this be explained?*

Response: This is a good point and we thank the reviewer for raising it. The relation shown in the left panel of Fig.3G is highly non-linear. The key message of this panel is that NPCs of lower SR (lower potency) generally have higher levels HES1 expression compared to high potency NPCs which (as the reviewer correctly points out) exhibit a much broader range of gene expression values. In our opinion, it is unreasonable to assume that any one neural stem cell marker (such as e.g. HES1) may be enough to provide a “clean” separation of NPCs into high and low potency. After all, cells are complex systems and we suspect that bona-fide NPCs (i.e. those which truly exhibit lower potency and are committed for differentiation into neural cell types) are those which highly express a minimum number of neural stem cell markers. This is certainly an important question to explore in future in collaboration with experimental groups, as this will require experimental verification.

Reviewer #3 (Remarks to the Author)

General Comment: Overall, this work is scientifically sound and well presented. The method proposed in this paper might contribute to the interpretation of single-cell experiments. Nevertheless, the availability of competing methods like monocle or DPT, and recent papers published on transcriptional fluctuation near lineage bifurcation points, might limit the impact of this manuscript. I would support publication in Nature Communications if three major issues are resolved.

Response: We thank the reviewer for taking time to evaluate our manuscript and for the overall positive feedback. We don't think that the availability of other methods like Monocle or DPT limits the impact of our work, since the main aim of these other methods is different. Algorithms such as Monocle or DPT are aimed at reconstructing cell-lineage trajectories. Signaling Entropy aims to quantify differentiation potency of single-cells. The application to reconstructing lineage trajectories is only one of many other applications. Moreover, an algorithm such as Monocle uses external information or prior biological knowledge (timepoint or expression of specific markers) to assign an identity to cells. The signaling

entropy measure proposed here does not use any external information to order cells in pseudotime or to assign them a level of differentiation potency. Furthermore, Monocle or DPT have not been applied or tested on cancer scRNA-Seq data. To the reviewer this might not seem as important, however, if we think of scenarios where prior biological knowledge maybe missing, for instance, if we want to identify novel progenitor subtypes for which no existing markers are known, or if we want to identify putative cancer stem cells, then a measure of potency which does not require external information is critical. Indeed, this is very much the reason why in this manuscript we present a potential application of signaling entropy to the identification of cancer stem cells (even in a non-invasive context (CTCs)). How could Monocle or DPT be used to identify putative cancer stem cells is unclear to us.

In the context of reconstructing cell-lineage trajectories, Signaling Entropy should be viewed as a complementary tool that could be used together with algorithms such as Monocle and DPT. Indeed, we showcase how it could be used in a time-course differentiation of human myoblasts. Although we believe that we made the complementary nature of Signaling Entropy clear in the manuscript, we now emphasize this point much more.

Major points:

Comment-1: While this approach, in particular the combination of the connectome with the transcriptome, is interesting I have a conceptual issue with this approach, which needs to be resolved before publication in Nature Communications. It relies on the idea of a monotonically decreasing entropy along the differentiation trajectory. Recently, two papers were published stating that differentiation decisions on the transcriptional level resemble "critical" transitions: A Richard et al., Plos Biol., (2016) and M Mojtahedi et al., Plos. Biol. (2016). While I am not entirely convinced that this conclusion follows rigorously from their observations, one of these papers (Richard et al.) reported an increase in transcriptional entropy shortly before the lineage decision is made, i.e. non monotonicity of transcriptional entropy. This is, at least superficially, in contradiction with the results reported here. While this might potentially be due to different definitions of the entropy, to avoid confusion in the field this discrepancy needs to be resolved or discussed in the manuscript.

Response: The reviewer has raised an excellent point and he is right in saying that the apparent discrepancy is mainly due to different definitions of "entropy". The "transcriptional entropy" of Richard et al is estimated for a single gene across single cells, and therefore reflects the amount of intercellular heterogeneity in the expression of a given gene. Our signaling entropy measure is estimated for a single-cell across genes in a large gene network, and is therefore genome-wide and cell-specific (Fig.1A-B). While the signaling entropy of single-cells will influence the amount of transcriptional heterogeneity and entropy as defined by Richard et al, the precise relation between the two entropies is non-trivial. Indeed, in panel Fig.1C, we show how we can assign single-cells into potency states, and we then define a Shannon Index (SI) or entropy over the whole cell population in terms of the distribution of potency states over single cells. This latter Shannon Index is more analogous to the transcriptional entropy of Richard et al. Indeed, as Fig.3C demonstrates, this Shannon Index is higher in a population of neural progenitor cells (NPCs) than in a population of hESCs. Thus, this Shannon Index has nothing to do with potency as such, i.e. it does not measure the average differentiation potency of single cells in a cell population. In contrast, our signaling entropy does measure potency of single cells in a cell population. Besides all of this, we should also note that Richard et al (as well as the study from Huang's lab) only studied a relatively small number of pre-selected genes, whereas our analysis is genome-wide.

To study the dynamic behavior of our Shannon Index in timecourse scRNA-Seq data will undoubtedly be an interesting future question to explore, but including such an analysis in this manuscript would only distract and extend this manuscript unnecessarily. In response to the reviewer's point, we have now included a new paragraph in Discussion, where we emphasize the differences between Signaling Entropy, the Shannon Index over a cell population and the transcriptional entropy from Richard et al.

Comment-2: Along the same lines, I am not convinced that, in principle, single-cell entropy should monotonically decrease with decreasing potency. One might equally well think that pluripotency or multipotency are tightly regulated attractor states, such that entropy increases only intermittently before a fate decision. I also found confusing the use of the word entropy in relation with the shallowness of Waddington's epigenetic landscape (Figure 1A). If the potential wells in this level of description are more shallow (have a larger second derivative) this would correspond to a higher plasticity (or susceptibility, in the terminology of statistical mechanics) but not necessarily to a higher entropy. Here, the terminology needs to be clarified in order to avoid confusion.

Response: We thank the reviewer for raising this point, which is indeed similar to the previous one, and our earlier response should address most of it. Briefly, our signaling entropy measure and the transcriptional entropy of Richard et al are different concepts and measures as explained above. However, we also feel that this reviewer may not be familiar with our previous signaling entropy work on bulk tissue (see Banerji CR et al Sci Rep 2013). There we provide very strong evidence that signaling entropy correlates with differentiation potency of cell-populations, within 3 distinct lineages (neural, mesenchymal and hematopoietic), and in timecourse differentiation and dedifferentiation experiments. For the convenience of the reviewer, we reproduce one of the main figures of that paper below:

Fig.3 from Banerji CR, Teschendorf AE Sci Rep 2013: (A) Multi-lineage analysis: Left panel: Comparison of normalised network entropy values of hESCs, hematopoietic stem cells (HSCs), T & B-cell lymphocytes plus natural killer cells (LYMPH/NKC), and monocytes plus neutrophils (MC/PMN). Middle panel:

Comparison of normalised network entropy values of hESCs, mesenchyma stem cells (MSCs) and differentiated osteoblasts (OST) and chondrocytes (CHO). Right panel: Comparison of normalised network entropy values of hESCs, neural stem cells (NSCs) derived from the hESCs, fetal neural stem cells (FNSC) and primary astrocytes (AC), as derived from the SCM compendium (Illumina arrays). Wilcoxon rank sum test P-values between consecutive groups in the differentiation hierarchy are given. (B) Dynamic changes in network entropy: Left panel: Network entropy changes in a time course de-differentiation and re-differentiation experiment of retinal pigment epithelium (RPE), with cell density indicating the initial plating density of RPE cells. Right panels: Network entropy rate (SR/maxSR, y-axis) changes of HL60 leukemic progenitor cells against time from initial stimulus with either ATRA or DMSO. The data points on the left indicate the less differentiated HL60 cells, whereas the ones on the far right represent differentiated neutrophils. We provide the R2 values and associated P-values from a linear regression.

We have now also added a new panel Fig.2H in the current manuscript which validates the signaling entropy concept in a developmental time course in the mouse lung epithelium. So, from an empirical perspective, Signaling Entropy does seem to discriminate both single cells and bulk samples according to their level of potency.

Does Signaling Entropy increase marginally before a transition to a cell-fate occurs, before eventually dropping? This is an interesting but also highly complex question, since the “time window” in which the hallmarks of criticality may be observable are quite short, and so, the limited time resolution of a differentiation experiment may not capture that “time window of criticality”. Nevertheless, we also feel that there might be some confusion regarding the different types of entropy that exist, as explained at length earlier. Our signaling entropy is a property of a single-cell, whereas the entropy concept that the reviewer is thinking of and which Richard et al use refers mainly to the entropy of the population of single cells. The reason for showing Waddington’s landscape is to make it clear to the reader that signaling entropy does provide a proxy to the height of attractors in this landscape. The relation of signaling entropy to the shallowness of the attractors is unexplored territory, which we plan to investigate in future.

In response to the reviewer’s excellent point, we now discuss some of these key points in Discussion.

Comment-3: Further to a comparison to the measures reported in the mentioned publications it would strengthen the impact of this work if the presented algorithm was compared to some of the methods that are already widely used in the field (like monocle or DPT). In how far does this method present an advantage in terms of (for example) accuracy, identification of branching points or scalability to established methods?

Response: We thank the reviewer for raising this point. We should clarify that methods like Monocle or DPT have as main aim to reconstruct cell-lineage trajectories, which we note is different to that of Signaling Entropy and SCENT: the main aim of Signaling Entropy (SCENT) is to quantify differentiation potency. Therefore, in our opinion, what is more meaningful is a comparison of Signaling Entropy to other proposed measures of differentiation potency, as for instance those which have been proposed as part of the SLICE and StemID algorithms. Indeed, we have now done so, and we reproduce below the table (which now forms Table-1 in our revised manuscript), which demonstrates that Signaling Entropy

compares very favourably to these other entropy-based potency measures in terms of the discriminative accuracy between cells that ought to be different in terms of differentiation potency:

Dataset		Signaling Entropy	SLICE	StemID
scRNA-Seq				
Chu1 (PI > NonPI)	P	3e-132	~1	3e-58
	AUC	0.96	<0.5	0.79
Chu2 (0h > 96h)	P	2e-38	0.94	1e-22
	AUC	0.97	<0.5	0.86
Trapnell (0h>72h)	P	6e-9	0.0003	2e-10
	AUC	0.74	0.65	0.75
Treutlein (E14>Adult)	P	5e-27	6e-26	5e-27
	AUC	1	0.998	1
Bulk RNA-Seq				
Chu3 (PI > NonPI)	P	4e-5	0.001	0.76
	AUC	0.99	0.90	<0.5

Table legend: Comparison of Signaling Entropy to SLICE and StemID as measures of differentiation potency in scRNA-Seq and bulk RNA-Seq datasets. Table lists one-tailed Wilcoxon rank sum test P-values and associated (one-tailed) AUCs, testing whether entropy is higher in the pluripotent or multipotent cells compared to the less potent cells in various scRNA-Seq and bulk RNA-Seq datasets. In Chu1, the comparison is between pluripotent (hESCs, n=374, PI) and non-pluripotent (n=644, NonPI) single cells. In Chu2, the comparison is between hESCs (0h, n=92) and definite endoderm progenitors sampled 96h later (n=188). In Trapnell, the comparison is between human myoblasts (0h, n=96) and differentiated skeletal muscle cells (72h, n=84). In Treutlein, the comparison is between early lung progenitors (E14, n=45) and mature alveolar cells (n=46). In Chu3, the comparison is between bulk hESCs (n=7) and non-pluripotent samples (n=12).

In Discussion, we also provide an explanation as to why Signaling Entropy is a more accurate and robust measure.

In response to the reviewer's point, we should also point that we do test SCENT on a human myoblast differentiation experiment (Fig.4), which is data also considered in the Trapnell et al Monocle paper. In fact, we state in Results that "Cells from this landmark which were present at differentiation induction exhibited intermediate potency expressing low levels of CDK1 (SI fig.S9 & Fig.4D), suggesting that these are "contaminating" interstitial mesenchymal cells that were already present at the start of the time course, in line with previous observations [Trapnell et al 2014]. Importantly, SCENT correctly predicts that the potency of all these mesenchymal cells in this landmark does not change during the time-course, consistent with the fact that these cells are not primed to differentiate into skeletal muscle cells, but which nevertheless aid the differentiation process [Trapnell et al 2014]." Thus, the output from SCENT is highly congruent with that of Monocle. One advantage of SCENT though is that it predicts the potency of these contaminating interstitial mesenchymal cells to be unaltered during the timecourse, which is expected biologically since these are not the cells that are differentiating. As far as we aware, neither Monocle or MPath (another algorithm tested on this data) are able to predict this particular feature.

On the other hand, we can't claim that SCENT is better than Monocle in reconstructing lineage trajectories: this would require at least 5-6 independent scRNA-Seq datasets where there is a clear "ground truth" which we note is lacking in most datasets. Given the large number of different methods now available (Monocle, DPT, Wanderlust, SCUBA, MPath, StemID, SLICE, SCORPIUS,...), and given that most of these have not been demonstrated to constitute an advance over others, a detailed comparison of all algorithms in terms of reconstructing cell-lineage trajectories is well beyond the scope and purpose of this manuscript. We should also emphasize yet again that the main component of SCENT is the use of Signaling Entropy as a means of quantifying differentiation potential of single cells. The subsequent application of Signaling Entropy to help infer reconstruct lineage trajectories merely reflects one possible application of Signaling Entropy. Our manuscript is about demonstrating the broad applicability of Signaling Entropy to the identification of stemness phenotypes in both normal and cancer scRNA-Seq data. As far as we are aware, no other measure or algorithm has been tested in both normal and cancer contexts.

Minor points:

- At several places the author reports very small p-values (e.g. $p < 1e-500$). Given the limited machine precision and the assumptions underlying the hypothesis testing I am skeptical that such small p-values can be estimated with certainty. Please clarify.

Response: Yes, the reviewer is right that P-values can't be estimated with such high precision. However, the "error" in the P-value estimate is insignificant, as $P < 1e-500$ indicates "strong statistical significance", a statement which is certainly correct. Bear in mind that we are comparing large numbers of cells, and hence the smallness of the P-values reflects the fact that we have large sample sizes. The issue brought up by the reviewer is well-known in the field, and conclusions are unaltered if we decide to replace $P < 1e-500$ with say $P < 1e-20$. However, we don't think that much is gained by doing so, as we think that the field is well-aware of this issue. Again, we feel that this is an editorial decision.

- Line 88: the term "connectivity profile" is unclear in the context of a correlation. A more explicit definition of the signalling entropy in the main text could help avoid confusion.

Response: We apologize for the lack of clarity. Connectivity profile means the connectivity or degree of the encoded proteins in the PPI network. However, this part of the paper has been modified anyway, as we have decided not to mention the Pearson correlation approximation in this manuscript. This is now only included in Online Methods.

- Line 227: I don't understand why Fig. 4a implies a stepwise reduction in signaling entropy. Couldn't the signaling entropy decrease smoothly between 0h and 24h?

Response: This is a typo. We have now rephrased this to "....., with a switch to lower entropy occurring at 24h".

- In general it would be helpful if terms like "transcriptomic heterogeneity" and "regulated heterogeneity" would be more precisely in the main text.

Response: We thank the reviewer for pointing this out. We now clarify the meaning of transcriptomic/expression heterogeneity in the Introduction. We now also clarify the term "regulated

heterogeneity” more precisely in the last subsection of Results: we now state “heterogeneity of cell populations is regulated in the sense that the transcriptomes of individual cells within the population differ in a manner which optimizes an objective function, such as pluripotency or homeostasis”. We note that we are using the term “regulated” in the sense proposed by MacArthur B & Lemishka Cell 2013, paper we cite when mentioning “regulated heterogeneity”.

- *Line 485: I could not find the definition of k_i*

Response: k_i is the degree or connectivity of gene i in the network.

Reviewers' comments:

Reviewer #1 (Remarks to the Author):

The authors have made some effort to improve the paper in response to the referees comments and I do think that it is improved. However, although I appreciate their efforts to improve the rationale for using signalling entropy, my main issues - (1) that the details of the mathematics of the method are not well presented and (2) that the method is actually much simpler than the paper would suggest - have remained mostly unaddressed.

The authors main idea: that "highly connected proteins are more likely to be highly expressed under conditions in which the requirement for a cell's phenotypic plasticity is high (e.g. as in a pluripotent state)" (lines 90-92) is interesting. Since it appears from their analysis that this may be true, this observation is potentially very important. However, the mathematical methods given in lines 526-553 are still not transparent: most readers of this paper will not know what terms such as "invariant measure", "detailed balance", "mean field approximation" etc. mean, and they are given without commentary or proper justification. These may well be useful notions but without a more thorough presentation of the mathematics, casual use of these terms unfortunately reads as an attempt to baffle the reader into thinking that something profound is being done. However, they indicate that the measure that ultimately arises from this process is indistinguishable in all the cases that they consider from a simple correlation. If this is the case, then it is still not clear to me why the extra complexity is needed. The reasons given in their response are not convincing: (1) entropy rate may have a useful meaning in the theory of random walks on networks but that does not mean that this notion is useful here: biologically what does a random walk on a protein-protein interaction network mean? (2) there may be hypothetical networks for which the correlation approximation is not good, but in all cases considered it is a good approximation, so this statement has to be taken on trust. (3) that signalling entropy has been used by the authors previously should not be a reason to continue using it here. (4) that signalling entropy can be used to extract important information not available from the correlation approximation could be a good reason to use it but no further details of how are given on how this might be done or what it might mean for the examples considered, so this also has to be taken on trust. (5) The final reason provided: "Finally, let us not forget that cells are complex systems, and differentiation potency is likely to be an "emergent" property. It would therefore be premature and wrong to conclude that the good performance of signaling entropy is entirely due to the correlation property." seems to be saying that there is a mysterious connection between "emergence" and "signalling entropy" but I do not really know what this statement means.

Overall I think that this is an interesting idea that is clearly practically useful. However I do not think that it has yet been demonstrated to be more effective, or better justified, than the much simpler method of taking a weighted average of expression with respect to the degree distribution of the network.

Reviewer #2 (Remarks to the Author):

The authors have addressed all my concerns in the revised version of the manuscript. I am now considering the manuscript suitable for publication in Nature Communications.

Reviewer #3 (Remarks to the Author):

In the revised version of the manuscript the authors addressed my issues regarding the first submission. Most of my criticism related to some of the terminology used throughout the manuscript. The manuscript has now been extended to include a clearer definition of the used

terms and a better differentiation from other concepts used in the field. I therefore support publication of the manuscript in Nature Communications.

Response to Reviewers' comments:

Reviewer #1:

General Comment: *The authors have made some effort to improve the paper in response to the referees comments and I do think that it is improved. However, although I appreciate their efforts to improve the rationale for using signalling entropy, my main issues - (1) that the details of the mathematics of the method are not well presented and (2) that the method is actually much simpler than the paper would suggest - have remained mostly unaddressed. Overall I think that this is an interesting idea that is clearly practically useful. However I do not think that it has yet been demonstrated to be more effective, or better justified, than the much simpler method of taking a weighted average of expression with respect to the degree distribution of the network.*

Response: We thank the reviewer for taking time to re-evaluate our manuscript and for the recognition that the manuscript has improved. Concerning the first point, it is not the purpose of a research manuscript to give an introduction to topics such as “Markov chains and Random Walks” where the definition and meaning of terms like “invariant measure” and “detailed balance” can be found. Nevertheless, in response to the reviewer’s point, we have now clarified the technical terminology in the Methods section. For instance, instead of “invariant measure” we now use the friendlier term of “steady-state probability”, which refers to the probability (in a steady state) of finding signaling flux (ie. the random walker) at a specific node in the network. Nodes with large steady-state probabilities thus mark genes that are highly influential in distributing signaling flux over the network. We hope that these changes have improved the clarity as much as it is possible and sensible within the scope of a research manuscript. Importantly, in the interest of transparency, we remind the reviewer that we have provided a user-friendly R-package, so that users can go through the code themselves, which will undoubtedly help them learn about the method.

With regards to the second point, we suspect that the reviewer is unfamiliar with our previous work, or is too busy and did not have time to read some of our previous papers on signaling entropy (see e.g. Banerji CR et al PLoS Comp Bio 2015, Sci Rep 2013, Teschendorff AE et al Methods 2014, Sci Rep 2015, West J et al Sci Rep 2012, Teschendorff AE et al BMC Systems Biology 2010). These previous studies (all performed on bulk tissue) are actually very important, because they demonstrate how specific biological information can be obtained from the signaling entropy framework, which would not have been possible using the Pearson correlation approximation. To appreciate this, it is important to first realize that (1) the global signaling entropy (i.e. calculated over the whole network) is estimated as a weighted average over local (i.e. for each gene/node in the network) signaling entropies (as shown in the equation in Fig.1B), and (2) that the local signaling entropies also contain valuable biological information. For instance, in West J et al Sci Rep 2012, we showed that the local signaling entropy of oncogenes is reduced in cancer compared to normal, whereas for tumor suppressors it is increased, in line with the fact that activation of an oncogene is associated with preferential activation of an oncogenic signaling pathway (thus a lower entropy). Another clear example can be found in Teschendorff AE et al Methods 2014, where we show that local signaling entropy correlates with resistance to drugs that target that particular gene. If we were to ditch the signaling entropy framework altogether, we would be restricting ourselves in the amount of biological information that could be

extracted. Thus, it makes absolutely no sense to us, to ditch signaling entropy in favor of an approximation which is less general and which offers a much more limited framework for biological interpretation.

Nevertheless, in response to the reviewer's point, we have now have added a new subsection in Results (and two new Supplementary Tables 3 & 4), which demonstrates that important biological information can indeed be extracted from the local signaling entropies in the single-cell setting. Briefly, what we demonstrate is that by ranking genes (nodes) according to their differential local entropy between pluripotent and non-pluripotent cells, that we can identify specific biological processes which underlie stemness. Top-ranked enriched biological terms included genes implicated in mRNA splicing and genes encoding mitochondrial ribosomal proteins (shown in Supplementary Tables 3-4). This is consistent with recent studies demonstrating that (i) mitochondrial activity influences the global transcription and splicing rate of cells (see e.g. *das Neves, R.P. et al. Connecting variability in global transcription rate to mitochondrial variability. PLoS Biol* **8**, e1000560 (2010), *Johnston, I.G. et al. Mitochondrial variability as a source of extrinsic cellular noise. PLoS Comput Biol* **8**, e1002416 (2012), *Guantes, R. et al. Global variability in gene expression and alternative splicing is modulated by mitochondrial content. Genome Res* **25**, 633-44 (2015)), and most importantly, that (ii) variations in such activity may influence stemness and differentiation (see e.g. *Schieke, S.M. et al. Mitochondrial metabolism modulates differentiation and teratoma formation capacity in mouse embryonic stem cells. J Biol Chem* **283**, 28506-12 (2008), *Wanet, A., Arnould, T., Najimi, M. & Renard, P. Connecting Mitochondria, Metabolism, and Stem Cell Fate. Stem Cells Dev* **24**, 1957-71 (2015), *Sukumar, M. et al. Mitochondrial Membrane Potential Identifies Cells with Enhanced Stemness for Cellular Therapy. Cell Metab* **23**, 63-76 (2016), *Hu, C. et al. Energy Metabolism Plays a Critical Role in Stem Cell Maintenance and Differentiation. Int J Mol Sci* **17**, 253 (2016), *Folmes, C.D. & Terzic, A. Energy metabolism in the acquisition and maintenance of stemness. Semin Cell Dev Biol* **52**, 68-75 (2016)). We would not have been able to retrieve this information using the Pearson correlation approximation since (i) this approximation does not care about the specific neighborhood interactions of a given gene, and (ii) it is a global measure which does not admit a local (gene-based) version.

Secondly, we point out that the Pearson correlation approximation is mentioned in the manuscript only because it provides valuable biological understanding and intuition as to why signaling entropy works. While we agree with the reviewer that the observation that “potency is encoded by the positive correlation between expression and connectivity” is an important one, it is equally important to realize that theoretically such a positive correlation is not sufficient for the signaling entropy to increase. To understand this, first note that the correlation between expression and connectivity constitutes an empirical approximation to signaling entropy, which holds only if certain assumptions are valid (as explained in Methods). One of these assumptions, which we acknowledge was not clearly spelled out in the original version of the manuscript, is that the local signaling entropies are also positively correlated with node connectivity. This is because strictly speaking the signaling entropy is approximated by the 3-way correlation between transcriptome, connectome and local entropies (as now explained in a new improved subsection in Methods). In fact, if hubs become overexpressed, then this leads to an increased global signaling entropy only if the local entropies of the hubs also increase. The figure below demonstrates (included as new SuppFig.7A) that there is indeed a positive correlation of differential local entropy (between pluripotent and non-pluripotent single cells) with node connectivity/degree, and that it is much more pronounced than that of differential gene expression and connectivity:

Figure legend: Left panel: Boxplot of the t -statistics of differential gene expression (y-axis) against gene/node connectivity (x-axis) with node degrees binned in groups as indicated. Pearson correlation coefficient and associated P -value are given. Red line is the least squares regression. Right panel: As left panel, but now for the t -statistics of differential local entropy (y-axis).

What this figure demonstrates is that the correlation between local entropies and connectivity is the “more profound” principle: hence, on the realistic PPI networks considered here, the empirical observation “that the Pearson correlation between expression and connectivity provides a very good proxy to signaling entropy and potency” is deeply embedded within the signaling entropy framework.

Thirdly, signaling entropy and its Pearson correlation approximation are not equivalent, as there could be networks where the two measures give very different answers. Indeed, further below (also shown in a new SuppFig.7D), we clearly demonstrate that the two measures can be anti-correlated if we alter some of the topological features of PPI networks.

Comment-1: The authors main idea: that “highly connected proteins are more likely to be highly expressed under conditions in which the requirement for a cell’s phenotypic plasticity is high (e.g. as in a pluripotent state)” (lines 90-92) is interesting. Since it appears from their analysis that this may be true, this observation is potentially very important.

Response: We should point out that the main idea is not what the reviewer is stating but in fact, it is the use of signaling entropy to estimate differentiation potency, as shown in Fig.1A-B. In fact, the main idea presented in the manuscript is the following: “Signaling entropy can be thought of as quantifying the overall level of signaling promiscuity of biological processes within the network. In effect, this quantifies the efficiency, or speed, with which signaling can diffuse over the whole network, and therefore measures the number of separate biological processes which are in some sense active. Since a committed, or differentiated cell, preferentially activates and deactivates specific processes (e.g. pathways) in the network, the expectation is that this would manifest itself as a lower entropy rate since signaling can’t diffuse to the regions of the network describing inactive processes”.

The other statement, i.e. “highly connected proteins are more likely to be highly expressed under conditions in which the requirement for a cell’s phenotypic plasticity is high (e.g. as in a pluripotent state)” is deeply embedded within the signaling entropy framework, and was only mentioned in order to help the reader gain some biological intuition. Given that this has caused confusion to this reviewer, we

have now restructured the first subsections of Results. We now provide an improved explanation of what signaling entropy is measuring, and we have a separate subsection to emphasize that signaling entropy can be approximated by the correlation between expression and connectivity (new SuppFig.7B-C), but that these are also non-equivalent measures (new SuppFig.7D).

Comment-2: However, the mathematical methods given in lines 526-553 are still not transparent: most readers of this paper will not know what terms such as "invariant measure", "detailed balance", "mean field approximation" etc. mean, and they are given without commentary or proper justification. These may well be useful notions but without a more thorough presentation of the mathematics, casual use of these terms unfortunately reads as an attempt to baffle the reader into thinking that something profound is being done.

Response: In response to this point, we now provide a brief explanation and definition of terms such as "invariant measure" and "detailed balance" in the relevant subsections in Methods. We have also improved the overall clarity of the assumptions underlying the derivation of the Pearson correlation approximation.

Comment-3: However, they indicate that the measure that ultimately arises from this process is indistinguishable in all the cases that they consider from a simple correlation. If this is the case, then it is still not clear to me why the extra complexity is needed. The reasons given in their response are not convincing: entropy rate may have a useful meaning in the theory of random walks on networks but that does not mean that this notion is useful here: biologically what does a random walk on a protein-protein interaction network mean?

Response: The reviewer is not right in stating that they are "indistinguishable". Once again, they are different measures and are not equivalent. Signaling entropy can be thought of as the efficiency or speed with which a random walker visits every node in the network. In our context, the "random walk" represents a probabilistic signaling process over the cellular interaction network, and what this allows us to do is in effect to quantify the number of separate biological processes which are in some sense "active". In a differentiated cell where specific pathways become activated whilst others become inactivated, the expectation would be that there is a bigger difference or spread in activity levels between these processes, thus leading to a lower signaling entropy, i.e it would take a random walker a much longer time to "explore" every corner in the network. This interpretation is not possible using a simple Pearson correlation approximation, because the correlation approximation does not care which nodes are neighbours of which other nodes, for instance, it does not care if hubs are connected to other hubs or if hubs are connected to low-degree nodes. The Pearson correlation only cares about the connectivity value. That is why the two measures are not equivalent. The Pearson correlation measure is only an approximation to the signaling entropy, which works well on the PPI networks considered here. We also provide a mathematical and biological rationale for why this approximation works well: if, under conditions that demand high phenotypic plasticity, hubs tend to be overexpressed, then this favours a high signaling entropy regime, because high expression of hubs allows the signaling processes to diffuse more efficiently throughout the network, assuming that the local signaling entropies of hubs also increase. (Many of these concepts are elaborated in detail in Teschendorff AE et al Sci Rep 2015).

Comment-4: there may be hypothetical networks for which the correlation approximation is not good, but in all cases considered it is a good approximation, so this statement has to be taken on trust.

Response: In response to this, we have simulated a type of network to demonstrate unequivocally that the two measures are NOT equivalent. We present the data in the figure below (also shown as new SuppFig.7D), which shows that signaling entropy and its Pearson correlation approximation are in fact anti-correlated on this particular type of network (see figure legend for detailed explanation):

Figure legend: Scatterplot of Signaling Entropy (SR, x-axis) against its Pearson Correlation Approximation (pccSR, y-axis) for 100 simulated networks (labeled by different symbols/letters). For each network there are two data points shown in two colors (red & black), representing different gene expression distributions. As we can see, the black data points correspond to an expression distribution where there is a positive correlation ($pccSR \sim 0.3$) between expression and connectivity, whereas the red data points correspond to an expression distribution where there is no correlation between expression and connectivity ($pccSR \sim 0$). However, for each of the 100 simulated networks, the signaling entropy (SR) is higher for the expression distribution that is uncorrelated with the degree (red points). Each of the 100 simulated networks contained 1010 nodes, with 10 nodes representing hubs, each with a connectivity of 100, with the rest of the 1000 nodes having low-degree (1-5 connections). Importantly, in these simulated networks the hubs are NOT connected to each other. They are only connected via intermediate low-degree nodes. In real PPI networks, hubs are generally connected to other hubs, and in this scenario (as shown in e.g. SuppFig.3C), there is excellent agreement between SR and pccSR.

Comment-5: that signalling entropy has been used by the authors previously should not be a reason to continue using it here. that signalling entropy can be used to extract important information not available from the correlation approximation could be a good reason to use it but no further details of how are given on how this might be done or what it might mean for the examples considered, so this also has to be taken on trust.

Response: In our previous works, we demonstrate very clearly how signaling entropy is able to provide insights, which a simple Pearson correlation approximation can not. To appreciate this, it is important to

make it clear from the outset that signaling entropy is a global measure, obtained by taking a weighted average over local signaling entropies, i.e. it is obtained by averaging signaling entropy values for each gene/protein in the network (as shown in the equation of Fig.1). The local signaling entropy of a gene depends on its neighborhood interactions, and we have previously shown that important biologically relevant information can be extracted from these local signaling entropies. For instance, in our previous paper Banerji CR et al PLoS Comp Bio 2015, we show how ranking genes according to the difference in local signaling entropy between good and poor outcome breast cancers allows us to identify genes that are enriched for stemness properties, consistent with the view that poor outcome breast cancers are characterized by a higher stemness-like signature. This result would not have been obtained had we only used the Pearson correlation approximation, since this approximation does not use explicit gene-gene interaction information. Another example is provided by our study Teschendorff AE et al Methods 2014, where we show that the local signaling entropy value of a gene correlates with resistance to drugs that target the corresponding gene. Again, this insight can't be obtained from using the Pearson correlation approximation, since the latter is a global measure which by definition does not admit a local (gene-based) version.

Nevertheless, in response to the reviewer's point, we have now have added a new subsection in Results (and two new Supplementary Tables 3 &4), which in effect demonstrate that important biological information can indeed be extracted from the local signaling entropies in the single-cell setting (see our detailed response above under the reviewer's General Comment).

Comment-6: The final reason provided: "Finally, let us not forget that cells are complex systems, and differentiation potency is likely to be an "emergent" property. It would therefore be premature and wrong to conclude that the good performance of signaling entropy is entirely due to the correlation property." seems to be saying that there is a mysterious connection between "emergence" and "signalling entropy" but I do not really know what this statement means.

Response: "Emergence" is a well-known term in Systems Biology and Complexity Science, and are very surprised that the term is a mystery to this reviewer. "Emergence" is a phenomenon whereby larger entities (e.g. a cell or cell population) arise through interactions among smaller or simpler entities (in our case, the expression levels of genes), such that the larger entities (ie. the cells) exhibit properties (e.g pluripotency) which the smaller/simpler entities (ie. genes) do not exhibit. So, our point is that the signaling entropy framework, because it is more general, that it could potentially offer an improved understanding of an "emergent" phenomenon such as e.g. pluripotency. A simple approximation derived from signaling entropy is unlikely to offer a framework for understanding the emergence of complex phenomena. If the reviewer is still confused, we can strongly recommend the perspective article by Ben MacArthur Cell 2013, which we cite in the introduction of our manuscript.

Finally, we feel that the reviewer is not giving us enough credit and is focusing on a rather minor detail concerning the "complexity of our model". We don't think that our model is complex, and would like to end by pointing out to the reviewer what are the main novel advances of our manuscript. These are:

- (1) We present an in-silico measure of differentiation potency at the single-cell level which works on both normal and cancer cells. To the best of our knowledge, this is the first study to present a single-cell measure that correlates with stemness in both normal and cancer contexts.

- (2) This opens us to critically important applications, namely to the identification of putative cancer stem-cells from e.g. circulating tumor cells. To the best of our knowledge, this is the first study to demonstrate the potential of a single-cell potency measure for the identification of cancer stem-cells.
- (3) Signaling entropy is a highly accurate measure of single-cell differentiation potency outperforming other entropy-based measures. Moreover, to the best of our knowledge, signaling entropy is the only measure to correlate with stemness at the single cell AND bulk tissue level.
- (4) To the best of our knowledge, our study is also one of the first to demonstrate a single-cell differentiation potency measure which correlates with potency independently of cell-cycle phase.
- (5) Signaling entropy provides a general framework in which to understand the differentiation potency of a cell population in terms of those of single-cells, providing strong evidence that cell-cell interactions are an important driver of potency at the bulk level.
- (6) We provide a number of biological and technical reasons why signaling entropy is such a robust measure of differentiation potency. One of the biological reasons is the observation that expression of hubs is increased in cells of higher phenotypic plasticity, a novel system-biological insight.

REVIEWERS' COMMENTS:

Reviewer #1 (Remarks to the Author):

The authors have worked hard to revise the paper, and the additional results do go some way to addressing my comments. The section in which it is shown that biologically relevant information can be found using their method that is not available from the approximation is particularly helpful; as is the section in which it is shown that correlation approximation is not always good (a very minor point: they show that the approximation works well for networks in which hubs do not connect preferentially. Such networks are known as disassortative, but this is not said explicitly. It seems as though assortativity coefficient of the network determines how well their approximation works). Overall although I still have questions about this method I think that they have shown it to be an interesting and valuable approach, and the authors should not be detained with any more unnecessary revisions.